# PerfectDou: Dominating DouDizhu with Perfect Information Distillation

**Guan Yang**[1][*]  **Minghuan Liu**[2][*]  **Weijun Hong**[1]
**Weinan Zhang**[2]  **Fei Fang**[3]  **Guangjun Zeng**[1]  **Yue Lin**[1][†]
[1] NetEase Games AI Lab, [2] Shanghai Jiao Tong University, [3] Carnegie Mellon University
{yangguan,gzzengguanjun,gzlinyue}@corp.netease.com,
{minghuanliu,wnzhang}@sjtu.edu.cn, feif@cs.cmu.edu

## Abstract

As a challenging multi-player card game, DouDizhu has recently drawn much attention for analyzing competition and collaboration in imperfect-information games. In this paper, we propose PerfectDou, a state-of-the-art DouDizhu AI system that dominates the game, in an actor-critic framework with a proposed technique named perfect information distillation. In detail, we adopt a perfect-training-imperfect-execution framework that allows the agents to utilize the global information to guide the training of the policies as if it is a perfect information game and the trained policies can be used to play the imperfect information game during the actual gameplay. To this end, we characterize card and game features for DouDizhu to represent the perfect and imperfect information. To train our system, we adopt proximal policy optimization with generalized advantage estimation in a parallel training paradigm. In experiments we show how and why PerfectDou beats all existing AI programs, and achieves state-of-the-art performance.

## 1 Introduction

With the fast development of Reinforcement Learning (RL), game AI has achieved great success in many types of games, including board games (e.g., Go [22], chess [21]), card games (e.g., Texas Hold'em [3], Mahjong [15]), and video games (e.g., Starcraft [26], Dota [1]). As one of the most popular card games in China, DouDizhu has not been studied in depth until very recently. In perfect-information games such as Go, agent can observe all the events occurred previously including initial hand of each agent and all agents' actions. In contrast, DouDizhu is an imperfect-information game with special structure, and an agent does not know other agents' initial hands but can observe all agents' actions. One challenge in DouDizhu is that it is a three-player game with both competition and collaboration: the two *Peasant* players need to cooperate as a team to compete with the third *Landlord* player. In addition, DouDizhu has a large action space that is hard to be abstracted for search-based methods [31].

Although various methods have been proposed for tackling these challenges [29, 10], they are either computationally expensive or far from optimal, and highly rely on abstractions with human knowledge [31]. Recently, Zha et al. [31] proposed DouZero, which applies simple Deep Monte-Carlo (DMC) method to learn the value function with pre-designed features and reward function. DouZero is regarded as the state-of-the-art (SoTA) AI system of DouDizhu for its superior performance and training efficiency compared with previous works.

---

[*]Equal contribution. Yang is responsible for the basic idea, system design and implementation details; Liu mainly contributes to the methodology, writing and experimental design. †Corresponding author. Project page at https://github.com/Netease-Games-AI-Lab-Guangzhou/PerfectDou/.

Unfortunately, we find DouZero still has severe limitations in many battle scenarios, which will be characterized in Section D.3. To establish a stronger and more robust DouDizhu bot, in this paper, we present a new AI system named PerfectDou, and show that it leads to significantly better performance than existing AI systems including DouZero. The name of our program follows the key technique we use – perfect information distillation. The proposed technique utilizes a Perfect-Training-Imperfect-Execution (PTIE) framework, a variant of the popular Centralized-Training-Decentralized-Execution (CTDE) paradigm in multi-agent RL literature [5, 11]. Namely, we feed perfect-information to the agent in the training phase to guide the training of the policy, and only imperfect-information can be used when deploying the learned policy for actual game play. Correspondingly, we further design the card and game features to represent the perfect and imperfect information. To train PerfectDou, we utilize Proximal Policy Optimization (PPO) [19] with Generalized Advantage Estimation (GAE) [18] by self-play in a distributed training system.

In experiments, we show PerfectDou beats all the existing DouDizhu AI systems and achieves the SoTA performance in a 10k-decks tournament; moreover, PerfectDou is the most training efficient, such that the number of samples required is an order of magnitude lower than the previous SoTA method; for application usage, PerfectDou can be deployed in online game environment due to its low inference time.

## 2 Preliminaries

**Imperfect-Information Extensive-Form Games.** An imperfect-information extensive-form (or tree-form) game can be described as a tuple $G = (\mathcal{P}, \mathcal{H}, \mathcal{Z}, \mathcal{A}, \mathcal{T}, \chi, \rho, r, \mathcal{I})$, where $\mathcal{P}$ denotes a finite set of *players*, $\mathcal{A}$ is a finite set of actions, and $\mathcal{H}$ is a finite set of *nodes* at which players can take actions and are similar to states in an RL problem. At a node $h \in \mathcal{H}$, $\chi : \mathcal{H} \rightarrow 2^{\mathcal{A}}$ is the action function that assigns to each node $h \in \mathcal{H}$ a set of possible actions, and $\rho : \mathcal{H} \rightarrow \mathcal{P}$ represents the unique acting player. An action $a \in A(h)$ that leads from $h$ to $h'$ is denoted by the successor function $\mathcal{T} : \mathcal{H} \times \mathcal{A} \rightarrow \mathcal{H}$ as $h' = \mathcal{T}(h, a)$. $\mathcal{Z} \subseteq \mathcal{H}$ are the sets of terminal nodes for which no actions are available. For each player $p \in \mathcal{P}$, there is a reward function $r_p \in r = r_1, r_2, \ldots, r_{|\mathcal{P}|} : \mathcal{Z} \rightarrow \mathbb{R}$. Furthermore, $\mathcal{I} = \{\mathcal{I}_p | p \in \mathcal{P}\}$ describes the information sets (infosets) in the game where $\mathcal{I}_p$ is a partition of all the nodes with acting player $p$. If two nodes $h, h'$ belong to the same infoset $I$ of player $p$, i.e., $h, h' \in I \in \mathcal{I}_p$, these two nodes are indistinguishable to $p$ and will share the same action set. We use $I(h)$ to denote the infoset of node $h$. Upon a certain infoset, a policy (or a behavior strategy) $\pi_p$ for player $p$ describes which action the player would take at each infoset. A policy can be stochastic, and we use $\pi_p(I)$ to denote the probability vector over player $p$'s available actions at infoset $I$. With a slight abuse of notation, we use $\pi_p(h)$ to denote the stochastic action player $p$ will take at node $h$. For two nodes $h, h'$ that belong to the same infoset $I$, it is clear that $\pi_p(h) = \pi_p(h') = \pi_p(I)$. Therefore, the objective for each player $p$ is to maximize its own total expected return at the end of the game: $R_p \triangleq \mathbb{E}_{Z \sim \pi}[r_p(Z)], Z \in \mathcal{Z}$.

**The Three-Players DouDizhu Game.** DouDizhu (a.k.a. Fight the Landload) is a three-player card game that is popular in China and is played by hundreds of millions of people. Among the three players, two of them are called the *Peasants*, and they need to cooperate as a team to compete against another player called the *Landlord*. The standard game consists of two phases, bidding and cardplay. The bidding phase designates the roles to the players and deals leftover cards to the *Landlord*. In the cardplay phase, the three players play cards in turn in clock-wise order. Within a game episode, there are several *rounds*, and each begins with one player showing a legal combination of cards (solo, pair, etc.). The subsequent players must either choose to pass or beat the previous hand by playing a more superior combination of cards, usually in the same category. The round continues until two consecutive players choose to pass and the player who played the last hand initiates to the next round. DouDizhu is in the genre of shedding where the player wins by emptying his's hand, or loses vice versa. Therefore, in this game, the suit does not matter but the rank does. The score of a game is calculated as the base score multiplied by a multiplier determined by specialized categories of cards (Appendix B.2 shows the details). In this paper, we only consider the cardplay phase, which can be formulated as an imperfect-information game. More detailed information about the game can be referred to [31].

The key challenges of DouDizhu include how the *Peasants* work as a team to beat the *Landlord* with card number advantage using only imperfect information. For example, one *Peasant* can try

to help his teammate to win by always trying the best to beat the *Landlord*'s cards and play cards in a category where the teammate has an advantage. In addition, the action space of DouDizhu is particularly large, and there are 27,472 possible combinations of cards that can be played in total with hundreds of legal actions in a hand. Furthermore, the action space cannot be easily abstracted since improperly playing a card may break other potential card combinations in the following rounds and lead to losing the game.

## 3 Methodology

In this section, we first introduce how perfect-training-imperfect-execution works for a general imperfect-information game. Then, we formulate the DouDizhu game as an imperfect-information game to solve.

### 3.1 Perfect Information Distillation

In card games as DouDizhu, the imperfect-information property comes from the fact that players do not show their hand cards to the others. And therefore the critical challenge for each player is to deal with the indistinguishable nodes from the same infoset. For such games, consider we can construct a strategically identical perfect-information game and allow one player to observe distinguishable nodes, then, the decisions at each node can rely on the global information and he may have more chances to win the game, like owning a cheating plug-in. This motivates us to utilize the distinguishable nodes for training the agents of imperfect-information games, and therefore we propose the technique of perfect information distillation.

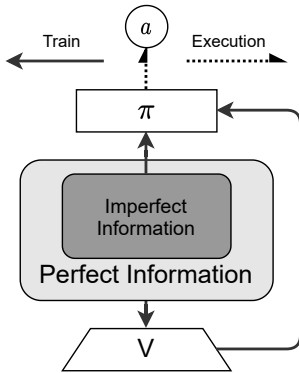

Figure 1: Overview of perfect information distillation within a perfect-training-imperfect-execution framework. The value network takes additional information (such as other players' cards in Poker games) as input, while the policy network does not.

In general, the perfect information distillation is a framework trained in perfect-training-imperfect-execution (PTIE) paradigm, a variant of centralized-training-decentralized-execution (CTDE) [5, 11], as illustrated in Fig. 1. Particularly, CTDE constructs the value function with all agents' observations and actions for general multi-agent tasks. By comparison, our proposed PTIE is designed for imperfect-information games where additional perfect information is introduced in the training stage. In this work, we consider to initiate PTIE with actor-critic [25] (but PTIE is not limited to actor-critic), which is a template of policy gradient (PG) methods, proposed in the RL literature towards maximizing the expected reward of the policy function through PG with a value function:

$$\nabla_{\theta_p} J = \mathbb{E}_\pi[\nabla_{\theta_p} \log \pi_{\theta_p}(a|s) Q_\pi(s,a)] , \tag{1}$$

where $s$ denotes a state in an RL problem, $Q$ is the state-action value function learned by a function approximator, usually called the critic. Notice that the critic is playing the role of evaluating how good an action is taken at a specific situation, but only at the training time. When the agent is deployed into inference, only the policy $\pi$ can be used to inferring feasible actions. Therefore, for imperfect-information games, we can provide additional information about the exact node the player is in to train the critic with self-play, as long as the actor does not take such information for decision making. Intuitively, we are distilling the perfect information into the imperfect policy.

Formally, for each node $h$, we construct a distinguishable node $D(h)$ for the strategically identical perfect-information game. Then, we define the value function at $D(h)$, $V_{\pi_p}(D(h)) = \mathbb{E}_{a \sim \pi_p, h^0=h}[Q_{\pi_p}(D(h), a)] = \mathbb{E}_{Z|\pi_p, h^0=h}[r_p(Z)]$ as the expected value of distinguishable nodes. In the sequel, we propose a simple extension of actor-critic policy gradient considering parameterized policy $\pi_{\theta_p}$ for each player $p$:

$$\nabla_{\theta_p} J = \mathbb{E}_{\pi_p}[\nabla_{\theta_p} \log \pi_{\theta_p}(a|h) Q_{\pi_p}(D(h), a)] . \tag{2}$$

In practice, we use a policy network (actor) to represent the policy $\pi_p$ for each player, which takes as input a vector describing the representation at an indistinguishable node $h$ that the player can observe

during the game. For estimating the critic, a value network is utilized, which takes representation of the global information at the distinguishable node $D(h)$. In other words, the value network takes additional information as inputs (such as other players' cards in Poker games) or targets (such as immediate rewards computed upon others' hands), while the policy network does not. During the training, the value function updates the values for all distinguishable nodes; then, it trains the policy on every node on the same infoset from sampled data, which implicitly gives an expected value estimation on each infoset. In practice, the generalization ability of neural networks enables the policy to find a better solution, which is also the advantage for using the proposed PTIE framework.

PTIE is a general way for training imperfect-information game agents using RL. In addition, the optimality point of independent RL in multi-agent learning is exactly one of an Nash equilibria (NE) and it has been proved that training RL algorithms with self-play can converge to an NE in two-player cases (although no convergence guarantees for three-player games) [9, 12], which may provide insights why PTIE works. We expect that with PTIE, players can leverage the perfect information during inference to derive coordination and strategic policies. In experiments, we show that this allows PerfectDou to cooperate with each other (as *Peasants*) or compete against the team (as the *Landlord*).

## 3.2 DouDizhu as An Imperfect-Information Game

As is mentioned in Section 2, the cardplay phase of DouDizhu can be regarded as an imperfect-information game with three players. At each node $h$, its infoset $I_p$ contains all combinations of the other's invisible handcards. Then at each level of the game tree, the three players take clockwise turns to choose an available action with policies $\pi_p$ depending on the infoset $I_p(h)$ of the current node $h$ with the reward function $r_p$. The path from the root of the game tree to a node $h$ contains the initial hand of all players and all the historical moves of all players. The reward functions at leaf nodes are set to be the score the players win or lose at the end of the game.

## 4 PerfectDou System Design

In this section, we explain how we construct our PerfectDou system in detail, with the proposed perfect information distillation technique, and several novel components designed for DouDizhu that help it summit the game. Particularly, PTIE requires different representations as input layer for the policy and the value network by feeding the value function with perfect information (distinguishable nodes) and the policy with imperfect information (indistinguishable nodes).

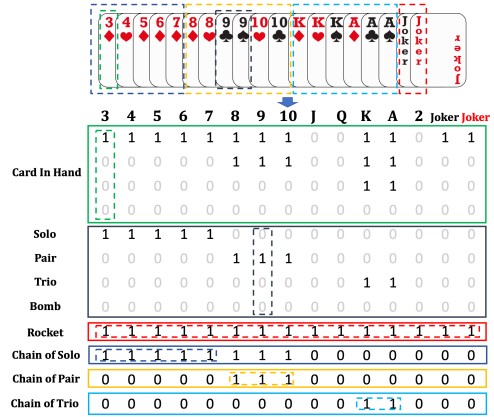

Figure 2: Card representation matrix. Columns stand for 15 different card ranks and rows stand for correspondingly designed features. The first 4 rows are the same as Zha et al. [31], and the last 8 rows are additional design for encoding the legal combination of cards.

### 4.1 Card Representation

In our system, we encode each feasible card combination with a $12 \times 15$ matrix, as shown in Fig. 2. Specifically, we first encode different ranks and numbers with a $4 \times 15$ matrix, where the columns correspond to the 15 ranks (including jokers) and similar to Zha et al. [31], the number of ones in the four rows of a single column represents the number of cards of that rank in the player's hand. Different from Zha et al. [31], we further propose to encode the legal combination of cards with the player's current hand, to help the agent realize the different property of various kinds of cards (see Appendix B.1). The feature sizes of each part are shown in Appendix C.1.

Table 1: Feature design of perfect-information (distinguishable nodes) and imperfect-information (indistinguishable nodes) for the game. Perfect features include all imperfect features.

| | FEATURE DESIGN | SIZE |
|---|---|---|
| IMPERFECT FEATURE | CURRENT PLAYER'S HAND | $1 \times 12 \times 15$ |
| | UNPLAYED CARDS | $1 \times 12 \times 15$ |
| | CURRENT PLAYER'S PLAYED CARDS | $1 \times 12 \times 15$ |
| | PREVIOUS PLAYER'S PLAYED CARDS | $1 \times 12 \times 15$ |
| | NEXT PLAYER'S PLAYED CARDS | $1 \times 12 \times 15$ |
| | 3 ADDITIONAL BOTTOM CARDS | $1 \times 12 \times 15$ |
| | LAST 15 MOVES | $15 \times 12 \times 15$ |
| | PREVIOUS PLAYER'S LAST MOVE | $1 \times 12 \times 15$ |
| | NEXT PLAYER'S LAST MOVE | $1 \times 12 \times 15$ |
| | MINIMUM PLAY-OUT STEPS OF HAND CARDS | 1 |
| | NUMBER OF CARDS IN CURRENT PLAYER'S HAND | 1 |
| | NUMBER OF CARDS IN PREVIOUS PLAYER'S HAND | 1 |
| | NUMBER OF CARDS IN NEXT PLAYER'S HAND | 1 |
| | NUMBER OF BOMBS | 1 |
| | FLAG OF GAME CONTROL BY CURRENT PLAYER | 1 |
| ADDITIONAL PERFECT FEATURE | PREVIOUS PLAYER'S HAND CARDS | $1 \times 12 \times 15$ |
| | NEXT PLAYER'S HAND CARDS | $1 \times 12 \times 15$ |
| | MINIMUM PLAY-OUT STEPS OF PREVIOUS PLAYER'S HAND CARDS | 1 |
| | MINIMUM PLAY-OUT STEPS OF NEXT PLAYER'S HAND CARDS | 1 |

## 4.2 Node Representation

In the game of DouDizhu, the distinguishable node $D(h)$ should cover all players' hand cards at $h$, along with the game and player status. Therefore, we propose to represent $h$ with imperfect features and $D(h)$ with perfect feature designs, shown in Tab. 1. In detail, the imperfect features include a flatten matrix[2] of $23 \times 12 \times 15$ and a game state array of $6 \times 1$. On the contrary, the perfect features consist of a flatten card matrix of $25 \times 12 \times 15$ and a game state array of $8 \times 1$. Therefore, they are totally asymmetric, and the imperfect features are a subset of the perfect features.

## 4.3 Network Structure and Action Representation

The PerfectDou system follows the general actor-critic design, and we take PPO [19] with GAE [18] as the learning algorithm. Slightly different from Eq. (2), PPO estimates the advantage $A_p = R_p - V_{\pi_p}$ as the critic instead of $Q_{\pi_p}$. For value network, we use an MLP to handle encoded features (the detailed structure is shown in Appendix C.3). As for the policy network, we first utilize an LSTM to encode all designed features; to encourage the agent to pay attention to specific card types, the proposed network structure will encode all the available actions into feature vectors, as depicted in Tab. 6. The output of the legal action probability is then computed with the action and game features, as illustrated in Fig. 3. Formally, we concatenate the node representation $e_s$ with each action representation $e_{a^i}$ separately, and get the legal action distribution:

$$p(a) = \text{softmax}(f([e_s, e_{a^i}]_{i=1}^N) , \tag{3}$$

where $a^i$ is the $i$-th action, $[\cdot]$ denotes the concatenation operation for $N$ available actions, and $f$ are layers of MLPs. This resembles the target attention mechanism in Ye et al. [28].

## 4.4 Perfect Reward Design

If we only care about the result at the end of the game, the reward at leaf nodes is rather sparse; in addition, players can only estimate their advantage of winning the game using imperfect information during the game, which could be inaccurate and fluctuated. Thanks to PTIE, we are allowed to impose an oracle reward function for DouDizhu at each node to enhance the perfect information modeled by the value function. In the training of PerfectDou, instead of estimating the advantage, we utilize an oracle[3] for evaluating each player, particularly, the minimum steps needed to play out all cards, which can be treated as a simple estimation of the distance to win. The reward function is

---

[2]Short for a matrix flattened to a one-dimensional vector.

[3]Implemented as a dynamic programming algorithm, see Appendix E for details.

then defined as the advantage difference computed by the relative distance to win of the two camps in two consecutive timesteps. Formally, at timestep $t$, the reward function is:

$$r_t = \begin{cases} -1.0 \times (\text{Adv}_t - \text{Adv}_{t-1}) \times l, & \text{Landlord} \\ 0.5 \times (\text{Adv}_t - \text{Adv}_{t-1}) \times l, & \text{Peasant} \end{cases} \tag{4}$$

$$\text{Adv}_t = N_t^{Landlord} - \min\left(N_t^{Peasant1}, N_t^{Peasant2}\right), \tag{5}$$

where $l$ is a scaling factor, and $N_t$ is the minimum steps to play out all cards at timestep $t$.

For instance, in a round, at timestamp $t$, the distance of the *Landlord* to win is 5 and the distances of two *Peasants* are 3 and 7, which means *Peasants* have a larger advantage since the relative distance is 2 for *Peasants* and -2 for the *Landlord*. However, if the *Landlord* plays a good hand such that both *Peasants* can not suppress, the *Landlord* will in result get a positive reward due to the decreased relative distance of the *Landlord*, i.e., from 2 to 1. Correspondingly, the *Peasants* would get a negative reward as their relative distances are getting larger. Such a reward function can encourage the cooperation between *Peasants*, since the winning distance is defined by the minimum steps of both players. In our implementation, the computation of the rewards is carried out after a round of the game, hence to promote training efficiency.

### 4.5 Distributed Training Details

To further expedite the training procedure, we design a distributed training system represented in Fig. 4. Specifically, the system contains a set of rollout workers for collecting the self-play experience data and sending it to a pool of GPUs; these GPUs asynchronously receive the data and store it into their local buffers. Then, each GPU learner randomly samples mini-batches from its own buffer and compute the gradient separately, which is then synchronously averaged across all GPUs and back propagated to update the neural networks. After each round of updating, new parameters are sent to every rollout worker. And each worker will load

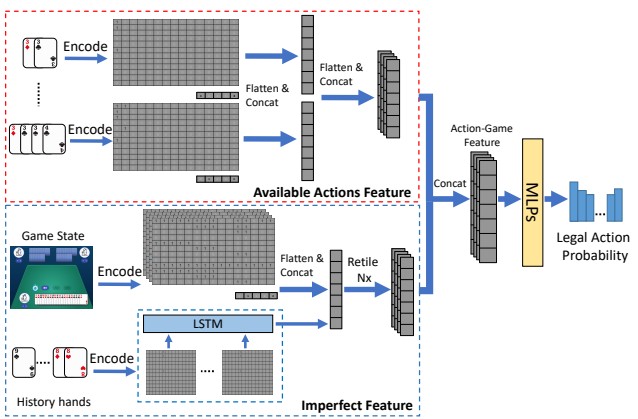

Figure 3: The policy network structure of PerfectDou system. The network predicts the action distribution given the current imperfect information of the game, including state information and available actions feature.

the latest model after 24 (8 for each player) steps sampling. Such a decoupled training-sampling structure will allow PerfectDou to be extended to large scale experiments. Our design of the distributed system borrows a lot from IMPALA [4], which also keeps a set of rollout workers to receive the updated model, interact with the environment and send back rollout trajectories to learners. The main difference is derived from the learning algorithm where we use PPO with GAE instead of actor-critic with V-trace [4]. Moreover, we keep three different models for *Landlord* and two *Peasants* separately which are only updated by their own data against the latest opponent models.

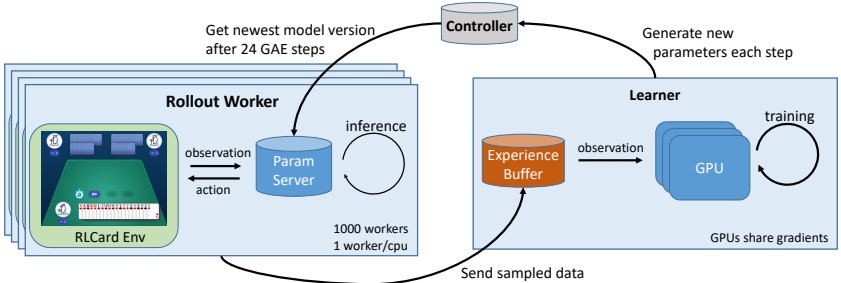

Figure 4: Illustration of the distributed training system.

Table 2: DouDizhu tournaments for existing AI programs by playing 10k randomly generated decks. Player A outperforms B if WP is larger than 0.5 or ADP is larger than 0 (highlighted in **boldface**). The algorithms are ranked according to the number of the other algorithms that they beat. We note that DouZero is the current SoTA DouDizhu bot and the gray rows highlight the comparison. Numerical results except marked ∗ are directly borrowed from Zha et al. [31].

| Rank | B \ A | PerfectDou | | DouZero | | DeltaDou | | RHCP-v2 | | CQN | | Random | |
|---|---|---|---|---|---|---|---|---|---|---|---|---|---|
| | | WP | ADP | WP | ADP | WP | ADP | WP | ADP | WP | ADP | WP | ADP |
| 1 | PerfectDou (Ours) | - | - | **0.543*** | **0.143*** | **0.584*** | **0.420*** | **0.543*** | **0.506*** | **0.862*** | **2.090*** | **0.994*** | **3.146*** |
| 2 | DouZero (Paper) | - | - | - | - | **0.586** | **0.258** | **0.764** | **1.671** | **0.810** | **1.685** | **0.989** | **3.036** |
| - | DouZero (Public) | 0.457* | -0.143* | - | - | **0.585*** | **0.253*** | **0.451*** | **0.060*** | **0.828*** | **1.950*** | **0.986*** | **3.050*** |
| 3 | DeltaDou | 0.416* | -0.420* | 0.414 | -0.258 | - | - | **0.691*** | **1.528*** | **0.784** | **1.534** | **0.992** | **3.099** |
| 4 | RHCP-v2 | 0.457* | -0.506* | **0.549*** | -0.060* | 0.309* | -1.423* | - | - | **0.770*** | **1.414*** | **0.990*** | **2.670*** |
| 5 | CQN | 0.138* | -2.090* | 0.190 | -1.685 | 0.216 | -1.534 | 0.230* | -1.414 * | - | - | **0.889** | **1.912** |
| 6 | Random | 0.006* | -3.146* | 0.011 | -3.036 | 0.008 | -3.099 | 0.010* | -2.670* | 0.111 | -1.912 | - | - |

# 5 Related Work

**Imperfect-Information Games.** Many popular card games are imperfect-information games and have attracted much attention. For instance, Li et al. [15] worked on the four-player game Mahjong and proposed a distributed RL algorithm combined with techniques like global reward prediction, oracle guiding, and run-time policy adaptation to win against most top human players; in addition, Lerer et al. [13] adopted search-based and imitation methods to learn the playing policy for Hanabi. Iterative algorithms such as Counterfactual Regret Minimization (CFR) and its variants [35, 17, 3] are also well-used for a fully competitive game, Hold'em Poker, whose action space is generally designed at the scale of tens (fold, call, check and kinds of bet) and the legal actions at each decision point are even less [17, 33]. In addition, CFR does not proven to converged to an Nash equilibrium in games with more than two players [14]. Consequently, they can be hardly designed for DouDizhu due to the large action space with difficult abstraction and the mixed game property (both cooperative and competitive, although there are few success cases for such a setting [16]).

**DouDizhu AI systems.** Besides the recent SoTA work DouZero [31], many researchers have made efforts on utilizing the power of RL into solving DouDizhu. However, simply applying RL algorithms such as DQN and A3C into the game can hardly make benefits [29]. Therefore, You et al. [29] proposed Combinational Q-Network (CQN) that reduces the action space by heuristics action decoupling; moreover, DeltaDou [10] utilized Monte-Carlo Tree Search (MCTS) for DouDizhu, along with Bayesian inference for the hidden information and a pre-trained kicker network for action abstraction. DeltaDou was also reported as reaching human-level performance. Zhang et al. [32] also relied on MCTS with predicting other players' actions. Recently, Zhao et al. [34] similarly proposed to model opponents' actions and train the policy model based on the DouZero architecture, reaching limited improvements. Nevertheless, as shown in this paper, we can instead distill such perfect-information knowledge like opponents' hand cards to the policy in a perfect-training-imperfect-execution style and reach a better performance.

# 6 Experiments

We conduct comprehensive experiments to investigate the following research questions. **RQ1**: How good is PerfectDou against SoTA DouDizhu AI? **RQ2**: What are the key ingredients of PerfectDou? **RQ3**: How is the inference efficiency of PerfectDou? To answer **RQ1**, we empirically evaluate the performance against existing DouDizhu programs. Regarding **RQ2**, we conduct ablation studies on key components in our design. And for **RQ3**, we calculate the average inference time for all algorithms involved. Finally, we conduct in-depth analysis, provide interesting case studies of PerfectDou. In the appendix, we report more results including a battle against skilled human players.

## 6.1 Experimental Setups

**Baselines.** We evaluate PerfectDou against the following algorithms under the open-source RL-Card Environment [30]: 1) **DouZero** [31]: A recent SoTA baseline method that had beaten every existing DouDizhu AI system using Deep Monte-Carlo algorithm. 2) **DeltaDou** [10]: An MCTS-based algorithm with Beyesian inference. It achieved comparable performance as human experts. 3) **Combinational Q-Network (CQN)** [29]: Based on card decomposition and Deep Q-Learning. 4)

**Rule-Based Algorithms**: Including the open-source heuristic-based program **RHCP-v2** [10, 31], the rule model in RLCard and a **Random** program with uniform legal moves. For evaluation, we directly take their public (or provided) codes and pre-trained models.

**Metrics.** The performance of DouDizhu are mainly quantified following the same metrics in previous researches [10, 31]. Specifically, given two algorithms A against B, we calculate: 1) **WP** (Winning Percentage): The proportion of winning by A in a number of games. 2) **ADP** (Average Difference in Points): The per-game averaged difference of scores between A and B. In other words, positive ADP means gaining scores while the negative represents losing it. This is a more reasonable metric for evaluating DouDizhu AI systems, because in real games players are evaluated by the scores obtained instead of their winning rates, as further discussed in Appendix B.2.

In our experiments, we choose ADP as the basic reward for all experiments training PerfectDou, which is augmented with the proposed reward signal in Section 4.4 during the training stage.

## 6.2 Comparative Evaluations

We conduct a tournament to demonstrate the advantage of our PerfectDou, where each pair of the algorithms play 10,000 decks, shown in Tab. 2 (**RQ1**). Since the bidding performance in each algorithm varies and poor bidding would affect game results significantly, for fair comparison, we omit the bidding phase and focus on the phase of cardplay. In detail, all games are randomly generated and each game would be played two times, i.e., each competing algorithm is assigned as *Landlord* or *Peasant* once. We use WP and ADP as the basic reward respectively for comparing over these two metrics for all evaluating methods.

Overall, PerfectDou dominates the leaderboard by beating all the existing AI programs, no matter rule-based or learning based algorithms, with significant advantage on both WP and ADP. Specifically, as noted that DouDizhu has a large variance where the initial hand cards can seriously determine the advantage of the game; even though, PerfectDou still consistently outperforms the current SoTA baseline – DouZero. However, we find that PerfectDou is worse than the result published in DouZero paper [31] when competing against RHCP. To verify this problem, we test the public model of DouZero (denoted as DouZero (Public) with grey color). To our surprise, its performance can match most of the reported results in their paper except the one against RHCP, where it only takes a WP of 0.452 lower than 0.5, indicating that the public model of DouZero can not beat RHCP as suggested in the original paper, and in fact PerfectDou is the better one.

It is also observed that some competition outcome has a high WP and a negative ADP. A potential reason can be explained as such agents are reckless to play out the bigger cards without considering the left hand, leading to winning many games of low score, but losing high score in the other games when their hand cards are not good enough. From our statistics of online human matches, the WP of winner is usually in a range of $0.52 \sim 0.55$ when the player tries to maximize its ADP.

We further reveal the sample efficiency of PerfectDou by comparing the competing performance w.r.t. different training steps. As shown in Tab. 3, we compare two versions of PerfectDou (1e9 and 2.5e9 steps) against two versions of DouZero (roughly 1e9 and 1e10 steps). From the tournament in Tab. 2, we know the final version of Per-

Table 3: Training efficiency comparison over 100k decks.

| A \ B | DouZero ($\sim$1e9) | | DouZero ($\sim$1e10) | |
|---|---|---|---|---|
| | WP | ADP | WP | ADP |
| PerfectDou (2.5e9) | - | - | 0.541 | 0.130 |
| PerfectDou (1e9) | 0.732 | 1.270 | 0.524 | 0.014 |
| DouZero ($\sim$1e10) | 0.698 | 1.150 | - | - |

fectDou (2.5e9 steps) outperforms the final version of DouZero ($\sim$1e10 steps). However, to our surprise, an early stage of PerfectDou is able to beat DouZero. With the same 1e9 training samples, PerfectDou wins DouZero with a large gap (WP of 0.732 and ADP of 1.270), which is even better than the 1e10 sample-trained DouZero. This indicates that PerfectDou is not only the best performance but also the most training efficient. The related training curves are shown in Appendix D.5.

## 6.3 Ablation Studies

We want to further investigate the key to the success of our AI system (**RQ2**). Specifically, we would like to analyse how our design of the feature and the training framework help PerfectDou dominate the tournament of DouDizhu. To this end, we evalu-

ate different variants of PerfectDou and the previous SoTA AI system – DouZero, including: a) *ImperfectDouZero*[4]: DouZero with our proposed imperfect-information features. b) *ImperfectDou*: PerfectDou with only imperfect-information features as inputs for the value function. c) *RewardlessDou*: PerfectDou without node reward. d) *Vanilla PPO*: Naive actor-critic training with imperfect-information features only and without additional reward. The ablation experiments are designed as competitions among ImperfectDou, RewardlessDou and PerfectDou against DouZero for comparing the effectiveness of perfect information distillation and perfect intermediate reward separately; while the

Table 4: Ablation studies over 100k decks.

| A \ B | DouZero (∼1e9) | | DouZero (∼1e10) | | ImperfectDouZero (∼1e9) | |
|---|---|---|---|---|---|---|
| | WP | ADP | WP | ADP | WP | ADP |
| PerfectDou (1e9) | 0.732 | 1.270 | 0.524 | 0.014 | 0.731 | 1.350 |
| ImperfectDou (1e9) | 0.717 | 1.180 | 0.486 | -0.057 | 0.723 | 1.320 |
| RewardlessDou (1e9) | 0.738 | 0.490 | 0.540 | -0.201 | 0.659 | 0.587 |
| Vanilla PPO(1e9) | 0.509 | -0.307 | 0.346 | -0.709 | 0.433 | -0.023 |

battle of ImperfectDouZero and DouZero against PerfectDou are designed for excluding the benefit from feature engineering. Results for all comparisons are shown in Tab. 4. Even with the imperfect features only, ImperfectDou can still easily beat DouZero with the same training steps; however, DouZero turns the tide with much more training data. Furthermore, our proposed node features seem not appropriate for DouZero to achieve a better results compared with its original design. Additionally, without the node reward, PerfectDou still beats DouZero with higher WP (in spite of sacrificing a lot of ADP), indicating the effectiveness of perfect reward in training, without which it would risk losing points to win one game. Finally, without both node reward and perfect feature design for the value function, vanilla PPO simply can not perform well. Therefore, we can conclude that our actor-critic based algorithm along with the PTIE training provides a high sample efficiency under our feature design, and the node reward benefits the rationality of our AI.

## 6.4 Runtime Analysis

We further conduct runtime analysis to show the efficiency of PerfectDou w.r.t. the inference time (**RQ3**), which is reported in Fig. 5. All evaluations are conducted on a single core of Intel(R) Xeon(R) Gold 6130 CPU @ 2.10GHz. The inference time of each AI could be attributed to its pipeline and implementation in the playing time. CQN uses a large Q network (nearly $10\times$ parameters larger than ours) with a complex card decomposer to derive reasonable hands. As a result, the inference time of CQN is the longest. Besides, both DeltaDou and RHCP-V2 contain lots of times of Monte

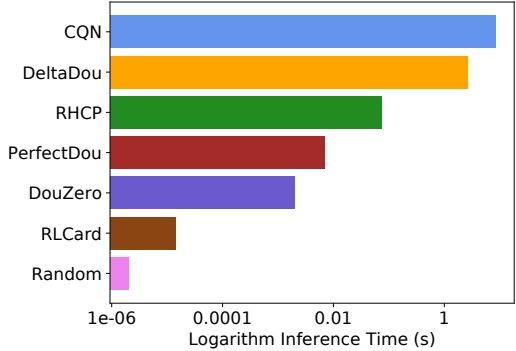

Figure 5: Comparison of the inference time.

Carlo simulations, thus slowing down the inference time. As comparisons, DouZero and Perfect-Dou only require one network forward inference time with a similar number of parameters. For RLCard, only handcraft rules are computed. Therefore, we can notice that PerfectDou is significantly faster than previous programs like DeltaDou, CQN and RHCP, yet is slightly slower than DouZero. To be more accurate, the average inference time of DouZero is 2 milliseconds compared with 6 milliseconds of PerfectDou. And the reason why PerfectDou is a bit slower than DouZero may due to the more complex feature processing procedure. Note that this can be further optimized in practice such as changing the implementation from Python to C++, which is common for AI deployment. In addition, the model inference time of DouZero we test is 1.9 ms while PerfectDou is 4 ms, the difference is slight. The above analysis suggests that PerfectDou is applicable and affordable to real-world applications such as advanced game AI.

## 6.5 In-depth Analysis and Case Studies

In our experiments, we find that DouZero is leaky and unreasonable in many battle scenarios, while PerfectDou performs better therein. To quantitatively evaluate whether PerfectDou is stronger and more reasonable, we conduct an in-depth analysis by collecting the statistics among the games, and

---

[4]Note that DouZero cannot acquire any perfect-information feature since it will play in a cheating style if so.

additionally illustrate some of the observations for qualitatively comparing the behavior of DouZero and PerfectDou. Detailed results can be further referred to Appendix D.2 and Appendix D.3, and here we list the key conclusions below:

The statistic analysis claims the rationality of PerfectDou: (i) when playing as the *Landlord*, PerfectDou plays fewer bombs to avoid losing scores and tends to control the game even when the *Peasants* play more bombs; (ii) when playing as the *Peasant*, two PerfectDou agents cooperate better with more bombs to reduce the control time of the *Landlord* and its chance to play bombs; (iii) when playing as the *Peasant*, the right-side *Peasant* agent (play after the *Landlord*) of PerfectDou throws more bombs to suppress the Landlord than DouZero, which is more like human strategy. From behavior observations, we also find: 1) DouZero is more aggressive but less thinking. 2) PerfectDou is better at guessing and suppressing. 3) PerfectDou is better at card combination. 4) PerfectDou is more calm. Beyond these, we also include battle results against skilled human players, additional training results and complete tournament results in the appendix.

## 7 Conclusion

In this paper, we propose PerfectDou, a SoTA DouDizhu AI system that dominates the game. PerfectDou takes the advantage of the perfect-training-imperfection-execution (PTIE) training paradigm, and is trained within a distributed training framework. In experiments we extensively investigate how and why PefectDou can achieve the SoTA performance by beating all existing AI programs with reasonable strategic actions. In fact, the PTIE paradigm is actually a variant of centralized-training-decentralized-execution (CTDE), applied for imperfect-information games in particular. Intuitively, PTIE is a general way for training imperfect-information game AI, with which we expect the value function can distill the perfect information to the policy which can only receive imperfect information. Although in this paper we only discuss its success on one of the hard poker games, DouDizhu, we believe it has the power to further improve the ability for other imperfect-information games, remaining a space of imagination to be explored by more future works.

## Acknowledgement

The SJTU team is partially supported by "New Generation of AI 2030" Major Project (2018AAA0100900) and National Natural Science Foundation of China (62076161). Minghuan Liu is also supported by Wu Wen Jun Honorary Doctoral Scholarship, AI Institute, SJTU. We thank Ming Zhou for helpful discussions on game theory, and anonymous reviewers for constructive suggestions.

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
