# Appendix

## A  Additional Related Work

**Train with global information, test with local one.**   Utilizing global information to reduce the complexity of imperfect-information games has also been investigated in some works. For example, AlphaStar [26], a grand-master level AI system for StarCraft II. In their implementation, the value network of the agent can observe the full information about the game state, including those that are hidden from the policy. They argue that such a training style improves training performance. In our work, we formulate the idea as Perfect-Training-Imperfect-Execution (PTIE) or perfect information distillation technique for imperfect-information games, and show the effectiveness on complicated card games like DouDizhu. Moreover, in Suphx [15], a strong Mahjong AI system, they used a similar method namely oracle guiding. Particularly, in the beginning of the training stage, all global information is utilized; then, as the training goes, the additional information would be dropped out slowly to none, and only the information that the agent is allowed to observe is reserved in the subsequent training stage. However, there are obvious difference between Suphx and PerfectDou. In Suphx, the perfect information is used by the actor and thus has to be dropped before the inference stage; on the contrary, PerfectDou feeds the critic with additional observations and distill the global information to the actor. Beyond games, Fang et al. [6] worked on trading for order execution and proposed a different technique using global information other than PTIE, which trained a student policy with imperfect (real) market information and policy distillation from a teacher policy trained with perfect (oracle) market information.

**Relation to sample-based CFR.**   CFR aims to minimize the total regret of policy by minimizing the cumulative counterfactual regret in each infoset. The definition of regret highly relates to the definition of advantage used in RL community, which has been shown in lots of previous works [23, 7]. Vanilla CFR [35] and many variants [27, 2] apply model-based approach to calculated all the weights of the game tree to update and obtain a good strategy (policy). However, when the game has long episodes and is hard for searching across the game tree, it is necessary to compute through trajectory samples, called sampled-based CFR methods [24, 8]. This resembles the learning procedure of RL algorithms. Recently, Fu et al. [7] proposed a new form of sample-based CFR algorithm, and shown that PPO is exactly a practical implementation of it (but not PTIE), revealing close connections between CFR and RL.

## B  More About DouDizhu

### B.1  Term of Categories

In the work of Zha et al. [31], they had shown a comprehensive introduction of DouDizhu game, so we think it may be wordy to repeat the stereotyped rules. However, for better understanding the cases shown in this paper, we introduce the typical term of categories in DouDizhu that are commonly used as follows. Note that all cards can suppress the cards in the same category with a higher rank, yet bomb can suppress any categories except the bomb with a higher rank. Rocket is the highest-rank bomb. **Kicker** refer to the unrelated or useless cards that players can deal out when playing some kind of categories of **main cards** (see below), which can be either a solo or a pair.

1. **Solo** : Any single card.
2. **Pair** : Two matching cards of equal rank.
3. **Trio** : Three individual cards of equal rank.
4. **Trio with Solo** : Three individual cards of equal rank with a Solo as the kicker.
5. **Trio with Pair** : Three individual cards of equal rank with a Pair as the kicker.
6. **Chain of Solo** : Five or more consecutive individual cards.
7. **Chain of Pair** : Three or more consecutive Pairs.
8. **Chain of Trio (Plane)** : Two or more consecutive Trios.
9. **Plane with Solo**: Two or more consecutive Trios with each has a distinct individual kicker card and Plane as the main cards.

10. **Quad with Solo** : Four-of-a-kind with two Solos as the kicker and Four-of-a-kind as the main cards.

11. **Quad with Pair** : Four-of-a-kind with a set of Pair as the kicker and Quad with Pair as the main cards.

12. **Quad with Pairs** : Four-of-a-kind with two sets of Pair as the kicker and Quad with Pairs as the main cards.

13. **Bomb** : Four-of-a-kind.

14. **Rocket** : Red and black jokers.

## B.2  Scoring Rules

In Zha et al. [31], they pay more attention to the win/lose result of the game but care less about the score. However, in real competitions, players must play for numbers of games and are ranked by the scores they win. And that is why we think ADP is a better metric for evaluating DouDizhu AI systems because a bad AI player can win a game with few scores but lose with much more scores.

Specifically, in each game, the *Landlord* and the *Peasants* have base scores of 2 and 1 respectively. When there is a bomb shown in a game, the score of each player doubles. For example, a *Peasant* player first shows a bomb of 4 and then the *Landlord* player suppresses it with a rocket, then the base score of each *Peasant* becomes 4 and the *Landlord* becomes 8. A player will win all his scores after winning the game, or loses all of them vice versa.

## C  Additional System Design Details

### C.1  Card Representation Details

In the system of PerfectDou, we augment the basic card in hand matrix with explicitly encoded card types as additional features, in order to allow the agent realizing the different properties of different kind of cards. The size details are shown in Tab. 5.

Table 5: Card representation design.

| CARD MATRIX FEATURE | SIZE |
|---|---|
| CARD IN HAND | $4 \times 15$ |
| SOLO | $1 \times 15$ |
| PAIR | $1 \times 15$ |
| TRIO | $1 \times 15$ |
| BOMB | $1 \times 15$ |
| ROCKET | $1 \times 15$ |
| CHAIN OF SOLO | $1 \times 15$ |
| CHAIN OF PAIR | $1 \times 15$ |
| CHAIN OF TRIO | $1 \times 15$ |

### C.2  Action Feature Details

Table 6: Action feature design.

| FEATURE DESIGN | SIZE |
|---|---|
| CARD MATRIX OF ACTION | $12 \times 15$ |
| IF THIS ACTION IS BOMB | 1 |
| IF THIS ACTION IS THE LARGEST ONE | 1 |
| IF THIS ACTION EQUALS THE NUMBER OF LEFT PLAYER'S CARDS IN HAND | 1 |
| IF THIS ACTION EQUALS THE NUMBER OF RIGHT PLAYER'S CARDS IN HAND | 1 |
| THE MINIMUM STEPS TO PLAY-OUT ALL LEFT CARDS AFTER THIS ACTION PLAYED | 1 |

The action features are a flatten matrix from $12 \times 15$ action card matrix plus $1 \times 6$ extra dimensions describing the property of the cards as shown in Tab. 6. Since the number of actions in each game state varies, which can lead to different lengths of action features, a fixed length matrix is flattened to store all action features where the non-available ones are marked as zero.

## C.3    Value Network Structure

The value network of PerfectDou is designed to evaluate the current situation of players, and we expect that the value function can utilize the global information, in other words, know the exact node the player is in. Therefore, we should feed additional information that the policy is not allowed to see in our design. Specifically, as shown in Fig. 6, the imperfect feature for indistinguishable nodes is encoded using the shared network as in the policy network; besides, we also encode the perfect feature of distinguishable nodes that the policy cannot observe during its game playing. The encoded vector are then concatenated to a simple MLP to get the scalar value output.

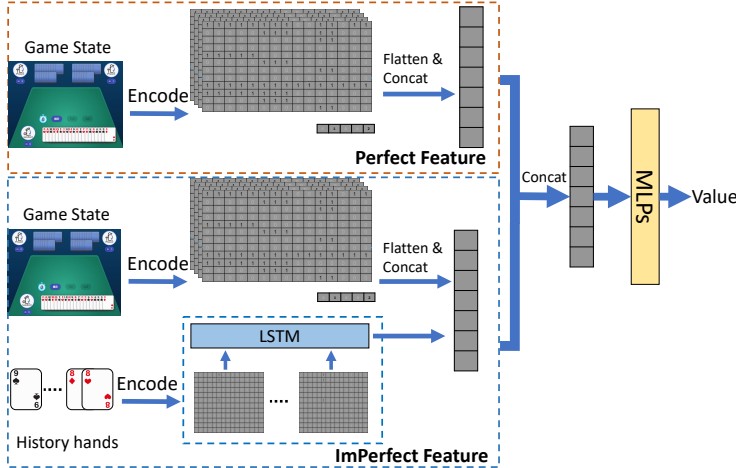

Figure 6: The value network structure of the proposed PerfectDou system. The network predicts values using both the imperfect feature and the perfect feature and distill the knowledge into the policy in the training.

## D    Experiments

### D.1    Setups, Hyperparameters and Training Details

In our implementation, a small distributed training cluster is built using 880 CPUs cores and 8 GPUs. Horovod [20] is used to synchronize gradients between GPUs, the total batch size is 1024, 128 for each GPU. The most important hyperparameters in our experiment are shown in Tab. 7. Specifically, in our design, we simplify the discrete action space from 27472 (include all possible combinations) into an abstract action space of 621 for learning the actor, followed by an decoding strategy to get the final action (see Appendix E.2 for more details).

During self-play training, we find a better practical solution for DouDizhu is to keep three different models for *Landlord* and two *Peasants* separately which is only updated by their own data against the latest opponent model. In the main training stage, the total reward function will be a basic reward (in this paper we use ADP all the time) augmented with the designed oracle reward as shown in Section 4.4, which is found to be extremely useful for accelerating convergence. In the final stage, the oracle reward is removed and only the ADP reward is used to fine-tune the model for reaching a better performance measured by the ADP metric.

### D.2    In-Depth Statistical Analysis

In our experiments, we find that DouZero is leaky and unreasonable in many battle scenarios, while PerfectDou performs better therein. To quantitatively evaluate whether PerfectDou is stronger and more reasonable, we conduct an in-depth analysis and collect the statistics among the games between DouZero and PerfectDou. Particularly, we organize games between PerfectDou and Douzero to play in different roles for 100,000 decks in each setting. Since the roles are assigned randomly instead of opting by agents themselves in our experiments, and the *Landlord* has a higher base score with three extra cards, we observe that playing as a *Landlord* is always harder to win and leads to negative ADPs. From the statistics shown in Tab. 8, we learn many lessons about the rationality of PerfectDou: (i) when playing as the *Landlord*, PerfectDou plays fewer bombs to avoid losing

Table 7: Hyperparameters. $^{*}$ refers to the maximum version gap allowed between the models used for sampling and training.

| | |
|---|---|
| Learning rate | 3e-4 |
| Optimizer | Adam |
| Discount factor $\gamma$ | 1.0 |
| $\lambda$ of GAE | 0.95 |
| Step of GAE | 24 (8 for each player) |
| Batch size | 1024 |
| Entropy weight of PPO | 0.1 |
| Length of LSTM | 15 (5 for each player) |
| Max model lag$^{*}$ | 1 |
| Intermediate reward scale | 50 |
| Policy MLP hidden sizes | [256, 256, 256, 512] |
| Policy MLP output size (action space size) | 621 |
| Value MLP hidden sizes | [256, 256, 256, 256] |
| Value MLP output size | 1 |

Table 8: Average per game statistics of important behaviors over 100k decks: `Game Len` is the average number of rounds in a game; `% Bomb` represents the average percentage of bombs (a type of card can suppress any categories except the bomb with a higher rank, see Appendix B) played in the game; `Left` and `Right` are the relative position to the *Landlord*; and `Landlord Control Time` measures the number of rounds that the landlord plays an action suppressing all other players.

| *Landlord* Agent | WP | ADP | Game Len | %Bomb of Left *Peasant* | %Bomb of *Landlord* | %Bomb of Right *Peasant* | Landlord Control Time | *Peasant* Agent |
|---|---|---|---|---|---|---|---|---|
| PerfectDou (2.5e9) | 0.446 | -0.407 | 33.347 | 68.05 | 28.46 | 74.90 | 12.993 | DouZero (∼1e10) |
| DouZero (∼1e10) | 0.421 | -0.461 | 33.911 | 66.24 | 28.73 | 75.29 | 9.005 | |
| PerfectDou (2.5e9) | 0.387 | -0.608 | 31.157 | 66.13 | 26.67 | 79.68 | 10.518 | PerfectDou (2.5e9) |
| DouZero (∼1e10) | 0.360 | -0.686 | 31.267 | 64.80 | 26.72 | 79.29 | 7.123 | |

scores and tends to control the game even the *Peasants* play more bombs; (ii) when playing as the *Peasant*, two PerfectDou agents cooperate better with more bombs to reduce the control time of the *Landlord* and its chance to play bombs; (iii) when playing as the *Peasant*, the right-side *Peasant* agent (play after the *Landlord*) of PerfectDou throws more bombs to suppress the Landlord than DouZero, which is more like human strategy.

### D.3 Case Study: Behavior of DouZero vs PerfectDou

In this section, we list some of the observations during the games for comparing the behavior of DouZero and PerfectDou to qualitatively support our analysis.

**DouZero is more aggressive but less thinking.** The first observation is that DouZero is extremely aggressive without considering the left hands. For instance, as shown in Fig. 7(a), in the beginning DouZero chooses a chain of solo but leaves the pair of 3, which can be dangerous since the pair of 3 is the one of the minimum cards and cannot suppress any card; Fig. 7(b) illustrates another strong case, where DouZero also chooses a chain of solo to suppress the opponent without considering the consequence of leaving a hand of solos. On the contrary, PerfectDou is more conservative and steady. We believe the proposed perfect information distillation mechanism helps PerfectDou to infer global information in a more reasonable way.

**PerfectDou is better at guessing and suppressing.** We observe another fact that the usage of perfect information distillation within the PTIE framework benefits PerfectDou a lot by suppressing the opponents in advance. In Fig. 7(c) shows a case when the teammate puts a pair of $T^{5}$, DouZero chooses to pass; on the contrary, PerfectDou chooses suppressing by a pair of $Q$ – the minimal pair of the *Landlord*.

**PerfectDou is better at card combination.** In the battle shown in Fig. 7(d), PerfectDou shows the better ability on the strategy of card combination. Specifically, PerfectDou chooses to split the plane (999, $TTT$ since it considers there is a chain of solo ($9TJQK$) left. However, DouZero only

---
$^{5}$We denote $T$(en) as the card 10 for simplicity.

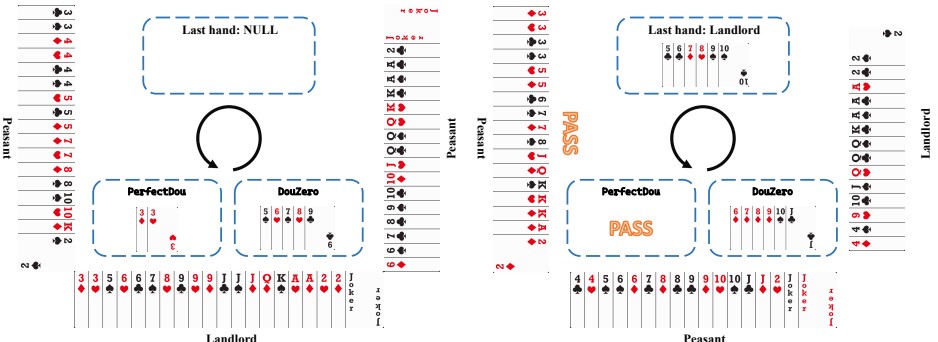

(a) Case study: DouZero is more aggressive by choosing a chain of solo in the beginning but leaves the pair of 3 in the hand.

(b) Case study: DouZero is more aggressive by suppressing the *Landlord* but less thinking on the consequence of the left hands of solos.

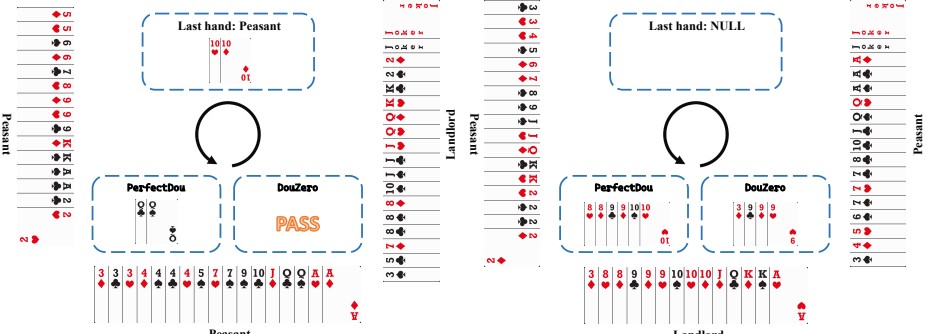

(c) Case study: the teammate shows a pair of T and DouZero chooses to pass; on the contrary, PerfectDou chooses suppressing by a pair of Q – the minimal pair of the opponent.

(d) Case study: PerfectDou chooses to split the plane (999, $TTT$) since it considers there is a chain of solo ($9TJQK$) left.

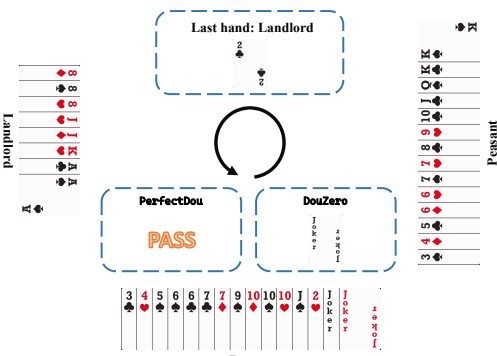

(e) Case study: DouZero splits the rocket bomb while PerfectDou chooses to keep it.

Figure 7: Case studies.

takes the trio, which will be easily suppressed by the opponent. This benefits from the proper design of the card representation and the action feature of PerfectDou.

**PerfectDou is more calm.** Fig. 7(e) depicts a typical and interesting scenario where PerfectDou shows its calm and careful consideration over the whole. In the game, the last hand is of the *Landlord* with a solo 2, and it only has 8 cards left in the hand. DouZero seems afraid and splits the rocket

bomb; however, PerfectDou benefits from the advantage reward design and is calm considering there is a greater chance on winning the game with a higher score by keeping the bomb.

## D.4  Battle Results Against Skilled Human Players

We further invite some skilled human players to play against PerfectDou. Particularly, each human player plays with two AI players. In other words, each game is involved with either two AI *Peasants* against one human *Landlord*, or one AI *Peasant* cooperating with one human *Peasant* against one AI *Landlord*. The results are shown in Tab. 9. One can easily observe that PerfectDou takes evident advantage during the game.

Table 9: Battle results against skilled human players for 1260 episodes of game.

| A \ B | Skilled human | |
|---|---|---|
|  | WP | ADP |
| PerfectDou (2.5e9) | 0.625 | 0.590 |

## D.5  Additional Training Results

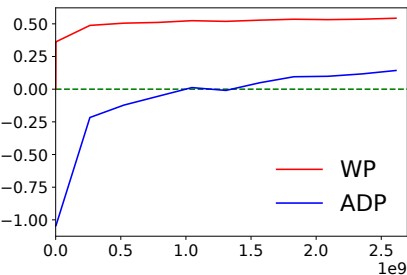

Figure 8: Learning curves of WP and ADP against the final model of DouZero w.r.t. timesteps for PerfectDou. Every evaluation contains 10,000 decks. PerfectDou is able to beat DouZero without considering the scores at the beginning of the training, around $1.5e6$ steps.

Fig. 8 shows the learning curves of WP and ADP against DouZero for PerfectDou with a single run, and every evaluation contains 10,000 decks. As shown in the figure, PerfectDou can easily beat DouZero (on WP) without considering the scores (ADP) at the beginning of the training; but after 1.5e9 steps of training, PerfectDou is able to fully beat DouZero (both WP and ADP).

## D.6  Extended Competition against DouZero

**Cycling Different Hands for the *Landlord*.**   In our main competition conducted in Section 6.2, all games are randomly generated and played twice with the same assigned hand cards for the *Landlord*, once each algorithm controlling the *Landlord* and once two *Peasants*. In this section, we test cycling the 3 hand cards of every randomly generated game for the *Landlord*, and test PerfectDou against DouZero by controlling the *Landlord* separately, which leads to 6 times of battle. In this situation, we obtain the results as follows showing that PerfectDou has consistent advantage (Tab. 10).

Table 10: Results of cycling different hands for Landlord by playing 100k decks.

| A \ B | PerfectDou | | DouZero | |
|---|---|---|---|---|
|  | WP | ADP |  |  |
| PerfectDou | - | - | 0.544 | 0.150 |
| DouZero | 0.456 | -0.150 | - | - |

***Peasants* Paired with Different AIs.**   We also include an interesting battle by pairing the peasants with different algorithms. Specifically, DouZero or PerfectDou plays the *Landlord* while one *Peasant* is assigned with DouZero and the other *Peasant* is played by PerfectDou. The results are concluded in Tab. 11, which reveal that when playing as the peasant, PerfectDou can better cooperate with its teammate than DouZero. And when playing as the landlord, PerfectDou outperforms DouZero against all types of opponents.

Table 11: Results of battles that the *Peasants* are paired with different AIs by 100k decks and 6 times of battle per deck. The results are evaluated from the *Landlord* side.

| Landlord \ Peasant | PerfectDou + DouZero | | PerfectDou | | DouZero | |
|---|---|---|---|---|---|---|
| | WP | ADP | WP | ADP | WP | ADP |
| PerfectDou | 0.424 | -0.448 | 0.389 | -0.606 | 0.452 | -0.375 |
| DouZero | 0.395 | -0.534 | 0.363 | -0.676 | 0.421 | -0.465 |

## D.7 Complete Tournament Results of ADP for *Landlord* and *Peasants*

We report the complete tournament results of ADP and WP for *Landlord* and *Peasants* in Tab. 12 and Tab. 13. PerfectDou tends to have more advantage of *Peasants* than that of *Landlord*, especially when competes against stronger baselines. We believe that the proposed perfect information distillation technique allows for better cooperation between two *Peasants*. In addition, since the roles are assigned instead of opting according to hand in our competition, and the *Landlord* has extra three cards and can lose a higher base score, the *Peasants* seems having more chance to win the game. Therefore, almost all methods can play better results as a *Peasant* than that as a *Landlord*.

Table 12: ADP results of DouDizhu tournaments for existing AI programs by playing 10k decks. L: ADP of A as Landlord; P: ADP of A as Peasants. Algorithm A outperforms B if the ADP of L or P is larger than 0 (highlighted in boldface). We note that DouZero is the current SoTA DouDizhu bot. Numerical results except marked $*$ are directly borrowed from Zha et al. [31].

| Rank | A \ B | PerfectDou | | DouZero | | DeltaDou | | RHCP-v2 | | CQN | | Random | |
|---|---|---|---|---|---|---|---|---|---|---|---|---|---|
| | | P | L | P | L | P | L | P | L | P | L | P | L |
| 1 | PerfectDou (Ours) | 0.656* | -0.656* | **0.686*** | -0.407* | **0.980*** | -0.145* | **0.872*** | **0.138*** | **2.020*** | **2.160*** | **3.008*** | **3.283*** |
| 2 | DouZero (Public) | 0.407* | -0.686* | 0.435* | -0.435* | **0.858*** | -0.342* | **0.166*** | -0.046* | **2.001*** | **1.368*** | **2.818*** | **3.254*** |
| 3 | DeltaDou | **0.145*** | -0.980* | **0.342*** | -0.858* | **0.476** | -0.476 | **1.878*** | **0.974*** | **1.849** | **1.218** | **2.930** | **3.268** |
| 4 | RHCP-v2 | -0.138* | -0.872* | **0.046*** | -0.166* | -0.974* | -1.878* | **0.182*** | -0.182* | **1.069*** | **1.758*** | **2.560*** | **2.780*** |
| 5 | CQN | -2.160* | -2.020* | -1.368* | -2.001* | -1.218 | -1.849 | -1.758* | -1.069* | **0.056** | -0.056 | **1.992** | **1.832** |
| 6 | Random | -3.283* | -3.008* | -3.254* | -2.818* | -3.268 | -2.930 | -2.780 | -2.560* | -1.832 | -1.991 | **0.883** | -0.883 |

Table 13: WP results of DouDizhu tournaments for existing AI programs by playing 10k decks. L: WP of A as Landlord; P: WP of A as Peasants. Algorithm A outperforms B if the WP of L or P is larger than 0.5 (highlighted in boldface). Numerical results except marked $*$ are directly borrowed from Zha et al. [31].

| Rank | A \ B | PerfectDou | | DouZero | | DeltaDou | | RHCP-v2 | | CQN | | Random | |
|---|---|---|---|---|---|---|---|---|---|---|---|---|---|
| | | P | L | P | L | P | L | P | L | P | L | P | L |
| 1 | PerfectDou (Ours) | **0.622*** | 0.378* | **0.640*** | 0.446* | **0.693*** | 0.474* | **0.609*** | 0.478* | **0.894*** | **0.830*** | **0.998*** | **0.990*** |
| 2 | DouZero (Public) | **0.554*** | 0.360* | **0.584*** | 0.416* | **0.684*** | 0.487* | 0.427* | 0.475* | **0.851*** | **0.769*** | **0.992*** | **0.986*** |
| 3 | DeltaDou | **0.526*** | 0.307* | **0.513*** | 0.317* | **0.588** | 0.412 | **0.768*** | **0.614*** | **0.835** | **0.733** | **0.996** | **0.987** |
| 4 | RHCP-v2 | **0.522*** | 0.391* | **0.525*** | **0.573*** | 0.386* | 0.232* | **0.536*** | 0.434* | **0.687*** | **0.853*** | **0.994*** | **0.985*** |
| 5 | CQN | 0.170* | 0.106* | 0.231* | 0.149* | 0.267 | 0.165 | 0.147* | 0.313* | 0.476 | **0.524** | **0.921** | **0.857** |
| 6 | Random | 0.010* | 0.002* | 0.014* | 0.008* | 0.013 | 0.004 | 0.015* | 0.006* | 0.143 | 0.080 | **0.654** | 0.346 |

# E  More Implementation Details

## E.1  The Oracle for Minimum Steps to Play Out All cards

In our paper, as mentioned in Section 4.4, we utilize an oracle for evaluating the minimum steps to play out all cards. Particularly, the oracle is implemented by a dynamic programming algorithm combined with depth-first-search, which can be referred to `https://www.cnblogs.com/SYCstudio/p/7628971.html` (which is also a competition problem of National Olympiad in Informatics in Provinces (NOIP) 2015). For completeness, we summarize the pseudocode for implementing such an algorithm in Algo. 1.

## E.2  Detailed Action Space

In our paper, we utilize a simplified discrete action space of 621 for learning the actor, since we observe that the original action space of 27472 (include all possible combinations) contains a large number of actions that can be abstract. For instance, actions like Bomb with kickers and Trio with kickers occupy the action space most due to the large number of combinations of kickers. To this end, we abstract actions with the same main cards into one action, and significantly reduce the action space. When the policy chooses an abstract action, we further deploy a simple decoding function to obtain the most preferred action in the original action space, as illustrated in Algo. 2. We note that this part is similar to an old implementation of RLCard [30]: `https://github.com/datamllab/rlcard/blob/d100952f144e4b0fd7186cc06e79ef277cda9722/rlcard/envs/doudizhu.py#L67`.

Below we list all the 621 discrete actions of PerfectDou in 15 categories, where the notation * in category (9) (10) (11) and (12) denotes the kicker.

(1) Solo (15 actions): 3, 4, 5, 6, 7, 8, 9, T[6], J, Q, K, A, 2, B, R

(2) Pair (13 actions): 33, 44, 55, 66, 77, 88, 99, TT, JJ, QQ, KK, AA, 22

(3) Trio (13 actions): 333, 444, 555, 666, 777, 888, 999, TTT, JJJ, QQQ, KKK, AAA, 222

(4) Trio with Solo (182 actions): 3334, 3335, 3336, 3337, 3338, 3339, 333T, 333J, 333Q, 333K, 333A, 3332, 333B, 333R, 3444, 4445, 4446, 4447, 4448, 4449, 444T, 444J, 444Q, 444K, 444A, 4442, 444B, 444R, 3555, 4555, 5556, 5557, 5558, 5559, 555T, 555J, 555Q, 555K, 555A, 5552, 555B, 555R, 3666, 4666, 5666, 6667, 6668, 6669, 666T, 666J, 666Q, 666K, 666A, 6662, 666B, 666R, 3777, 4777, 5777, 6777, 7778, 7779, 777T, 777J, 777Q, 777K, 777A, 7772, 777B, 777R, 3888, 4888, 5888, 6888, 7888, 8889, 888T, 888J, 888Q, 888K, 888A, 8882, 888B, 888R, 3999, 4999, 5999, 6999, 7999, 8999, 999T, 999J, 999Q, 999K, 999A, 9992, 999B, 999R, 3TTT, 4TTT, 5TTT, 6TTT, 7TTT, 8TTT, 9TTT, TTTJ, TTTQ, TTTK, TTTA, TTT2, TTTB, TTTR, 3JJJ, 4JJJ, 5JJJ, 6JJJ, 7JJJ, 8JJJ, 9JJJ, TJJJ, JJJQ, JJJK, JJJA, JJJ2, JJJB, JJJR, 3QQQ, 4QQQ, 5QQQ, 6QQQ, 7QQQ, 8QQQ, 9QQQ, TQQQ, JQQQ, QQQK, QQQA, QQQ2, QQQB, QQQR, 3KKK, 4KKK, 5KKK, 6KKK, 7KKK, 8KKK, 9KKK, TKKK, JKKK, QKKK, KKKA, KKK2, KKKB, KKKR, 3AAA, 4AAA, 5AAA, 6AAA, 7AAA, 8AAA, 9AAA, TAAA, JAAA, QAAA, KAAA, AAA2, AAAB, AAAR, 3222, 4222, 5222, 6222, 7222, 8222, 9222, T222, J222, Q222, K222, A222, 222B, 222R

(5) Trio with Pair (156 actions): 33344, 33355, 33366, 33377, 33388, 33399, 333TT, 333JJ, 333QQ, 333KK, 333AA, 33322, 33444, 44455, 44466, 44477, 44488, 44499, 444TT, 444JJ, 444QQ, 444KK, 444AA, 44422, 33555, 44555, 55566, 55577, 55588, 55599, 555TT, 555JJ, 555QQ, 555KK, 555AA, 55522, 33666, 44666, 55666, 66677, 66688, 66699, 666TT, 666JJ, 666QQ, 666KK, 666AA, 66622, 33777, 44777, 55777, 66777, 77788, 77799, 777TT, 777JJ, 777QQ, 777KK, 777AA, 77722, 33888, 44888, 55888, 66888, 77888, 88899, 888TT, 888JJ, 888QQ, 888KK, 888AA, 88822, 33999, 44999, 55999, 66999, 77999, 88999, 999TT, 999JJ, 999QQ, 999KK, 999AA, 99922, 33TTT, 44TTT, 55TTT, 66TTT, 77TTT, 88TTT, 99TTT, TTTJJ, TTTQQ, TTTKK, TTTAA, TTT22, 33JJJ, 44JJJ, 55JJJ, 66JJJ, 77JJJ, 88JJJ, 99JJJ, TTJJJ, JJJQQ, JJJKK, JJJAA, JJJ22, 33QQQ, 44QQQ, 55QQQ, 66QQQ, 77QQQ, 88QQQ, 99QQQ, TTQQQ, JJQQQ, QQQKK, QQQAA, QQQ22, 33KKK, 44KKK, 55KKK, 66KKK, 77KKK, 88KKK, 99KKK, TTKKK, JJKKK, QQKKK, KKKAA, KKK22, 33AAA, 44AAA, 55AAA, 66AAA, 77AAA,

---

[6]T for Ten (10).

**Algorithm 1** Calculate Minimum Step to Play Out All Cards

---

1: **function** MAIN
2:     $N_1 \leftarrow$ all possible number of single card
3:     $N_2 \leftarrow$ all possible number of pair card
4:     $N_3 \leftarrow$ all possible number of trio card
5:     $N_4 \leftarrow$ all possible number of bomb card
6:     **function** INITMATRIX($F$)
7:         **for** $action \in \{$***Solo***, ***Pair***, ***Trio***, ***Bomb***, ***Chain-of-Trio***, ***Trio-with-Pair***, ***Quad-with-Solos***, ***Quad-with-Pair***, ***Quad-with-Pairs***$\}$ **do**
8:             $d_1 \leftarrow$ the number of ***Solo*** card in $action$
9:             $d_2 \leftarrow$ the number of ***Pair*** card in $action$
10:             $d_3 \leftarrow$ the number of ***Trio*** card in $action$
11:             $d_4 \leftarrow$ the number of ***Bomb*** card in $action$
12:             $F[N_1, N_2, N_3, N_4] \leftarrow \min(F[N_1, N_2, N_3, N_4], F[N_1 - d_1, N_2 - d_2, N_3 - d_3, N_4 - d_4] + 1)$
13:             **if** $action$ is ***Trio*** **then**
14:                 $F[N_1, N_2, N_3, N_4] \leftarrow \min(F[N_1, N_2, N_3, N_4], F[N_1 + 1, N_2 + 2, N_3 - 1, N_4])$
15:             **end if**
16:             **if** $action$ is ***Bomb*** **then**
17:                 $F[N_1, N_2, N_3, N_4] \leftarrow \min(F[N_1, N_2, N_3, N_4], F[N_1 + 1, N_2 + 2, N_3 - 1, N_4])$
18:             **end if**
19:         **end for**
20:     **end function**
21:     **function** NOWSTEP($Cards$)
22:         **if** ***Rocket*** $\in Cards$ **then**
23:             $left\_cards \leftarrow$ left cards after playing out ***Rocket***
24:             **return** $\min($ NowStep ( $left\_cards$ ) $+ 1$, NowStep ( $left\_cards$ ) $+ 2$ )
25:         **end if**
26:         **if** only one ***Joker*** $\in Cards$ **then**
27:             $left\_cards \leftarrow$ left cards after playing out ***Joker***
28:             **return** NowStep ( $left\_cards$ ) $+ 1$
29:         **end if**
30:         **if** no ***Joker*** $\in Cards$ **then**
31:             calculate number of ***Solo*** $N_1$, ***Pair*** $N_2$, ***Trio*** $N_3$ and ***Bomb*** $N_4$ of $Cards$
32:             **return** $F[N_1, N_2, N_3, N_4]$
33:         **end if**
34:     **end function**
35:     **function** DFS($step$, $ans$, $Cards$)
36:         **if** $step > ans$ **then**
37:             **return** $ans$
38:         **end if**
39:         $ans \leftarrow \min($ $step$, NowStep($Cards$) )
40:         **for** ***Chain-of-Solo*** $\in Cards$ **do**
41:             $left\_cards \leftarrow$ left cards after playing out ***Chain-of-Solo***
42:             DFS($step + 1$, $ans$, $left\_cards$ of $Cards$)
43:         **end for**
44:         **for** ***Chain-of-Pair*** in $Cards$ **do**
45:             $left\_cards \leftarrow$ left cards after playing out ***Chain-of-Pair***
46:             DFS($step + 1$, $ans$, $left\_cards$ of $Cards$)
47:         **end for**
48:         **for** ***Plane-with-Solo*** $\in Cards$ **do**
49:             $left\_cards \leftarrow$ left cards after playing out ***Plane-with-Solo***
50:             DFS($step + 1$, $ans$, $left\_cards$ of $Cards$)
51:         **end for**
52:     **end function**
53:     Create a matrix $F$ of size $[N_1, N_2, N_3, N_4]$
54:     InitMatrix($F$)
55:     $step \leftarrow 0$ ,$ans \leftarrow +\infty$, $Cards \leftarrow$ all $Cards$ to be calculated
56:     DFS( $0$, $ans$, $Cards$ )
57: **end function**

---

**Algorithm 2** Decode action

---

1: **function** DECODE($M$)
2:     $A \leftarrow$ get all available actions from current hand
3:     $K \leftarrow$ get all kickers using the main card $M$ from $A$
4:     **for** $k$ in $K$ **do**
5:         calculate score $s$ of each $k$
6:         $N \leftarrow$ number of actions contains $k$ in $A$, $rank_k \leftarrow$ card rank of $k$
7:         $s \leftarrow 1.0 * N + 0.1 * rank_k$
8:     **end for**
9:     **return** $k$ with minimum $s$
10: **end function**

---

88AAA, 99AAA, TTAAA, JJAAA, QQAAA, KKAAA, AAA22, 33222, 44222, 55222, 66222, 77222, 88222, 99222, TT222, JJ222, QQ222, KK222, AA222

(6) Chain of Solo (36 actions): 34567, 45678, 56789, 6789T, 789TJ, 89TJQ, 9TJQK, TJQKA, 345678, 456789, 56789T, 6789TJ, 789TJQ, 89TJQK, 9TJQKA, 3456789, 456789T, 56789TJ, 6789TJQ, 789TJQK, 89TJQKA, 3456789T, 456789TJ, 56789TJQ, 6789TJQK, 789TJQKA, 3456789TJ, 456789TJQ, 56789TJQK, 6789TJQKA, 3456789TJQ, 456789TJQK, 56789TJQKA, 3456789TJQK, 456789TJQKA, 3456789TJQKA

(7) Chain of Pair (52 actions): 334455, 445566, 556677, 667788, 778899, 8899TT, 99TTJJ, TTJJQQ, JJQQKK, QQKKAA, 33445566, 44556677, 55667788, 66778899, 778899TT, 8899TTJJ, 99TTJJQQ, TTJJQQKK, JJQQKKAA, 3344556677, 4455667788, 5566778899, 66778899TT, 778899TTJJ, 8899TTJJQQ, 99TTJJQQKK, TTJJQQKKAA, 334455667788, 445566778899, 5566778899TT, 66778899TTJJ, 778899TTJJQQ, 8899TTJJQQKK, 99TTJJQQKKAA, 33445566778899, 445566778899TT, 5566778899TTJJ, 66778899TTJJQQ, 778899TTJJQQKK, 8899TTJJQQKKAA, 33445566778899TT, 445566778899TTJJ, 5566778899TTJJQQ, 66778899TTJJQQKK, 778899TTJJQQKKAA, 33445566778899TTJJ, 445566778899TTJJQQ, 5566778899TTJJQQKK, 66778899TTJJQQKKAA, 33445566778899TTJJQQ, 445566778899TTJJQQKK, 5566778899TTJJQQKKAA

(8) Chain of Trio (45 actions): 333444, 444555, 555666, 666777, 777888, 888999, 999TTT, TTTJJJ, JJJQQQ, QQQKKK, KKKAAA, 333444555, 444555666, 555666777, 666777888, 777888999, 888999TTT, 999TTTJJJ, TTTJJJQQQ, JJJQQQKKK, QQQKKKAAA, 333444555666, 444555666777, 555666777888, 666777888999, 777888999TTT, 888999TTTJJJ, 999TT-TJJJQQQ, TTTJJJQQQKKK, JJJQQQKKKAAA, 333444555666777, 444555666777888, 555666777888999, 666777888999TTT, 777888999TTTJJJ, 888999TTTJJJQQQ, 999TT-TJJJQQQKKK, TTTJJJQQQKKKAAA, 333444555666777888, 444555666777888999, 555666777888999TTT, 666777888999TTTJJJ, 777888999TTTJJJQQQ, 888999TT-TJJJQQQKKK, 999TTTJJJQQQKKKAAA

(9) Plane with Solo (38 actions): 333444**, 444555**, 555666**, 666777**, 777888**, 888999**, 999TTT**, TTTJJJ**, JJJQQQ**, QQQKKK**, KKKAAA**, 333444555***, 444555666***, 555666777***, 666777888***, 777888999***, 888999TTT***, 999TTTJJJ***, TTTJJJQQQ***, JJJQQQKKK***, QQQKKKAAA***, 333444555666****, 444555666777****, 555666777888****, 666777888999****, 777888999TTT****, 888999TTTJJJ****, 999TT-TJJJQQQ****, TTTJJJQQQKKK****, JJJQQQKKKAAA****, 333444555666777*****, 444555666777888*****, 555666777888999*****, 666777888999TTT*****, 777888999TT-TJJJ*****, 888999TTTJJJQQQ*****, 999TTTJJJQQQKKK*****, TTTJJJQQQKKKAAA*****

(10) Plane with Pair (30 actions): 333444****, 444555****, 555666****, 666777****, 777888****, 888999****, 999TTT****, TTTJJJ****, JJJQQQ****, QQQKKK****, KKKAAA****, 333444555******, 444555666******, 555666777******, 666777888******, 777888999******, 888999TTT******, 999TTTJJJ******, TTTJJJQQQ******, JJJQQQKKK******, QQQKKKAAA******, 333444555666********, 444555666777********, 555666777888********, 666777888999********, 777888999TTT********, 888999TTTJJJ********, 999TTTJJJQQQ********, TTTJJJQQQKKK********, JJJQQQKKKAAA********

(11) Quad with Solo (13 actions): 3333**, 4444**, 5555**, 6666**, 7777**, 8888**, 9999**, TTTT**, JJJJ**, QQQQ**, KKKK**, AAAA**, 2222**

(12) Quad with Pair (13 actions): 3333****, 4444****, 5555****, 6666****, 7777****, 8888****, 9999****, TTTT****, JJJJ****, QQQQ****, KKKK****, AAAA****, 2222****

(13) Bomb (13 actions): 3333, 4444, 5555, 6666, 7777, 8888, 9999, TTTT, JJJJ, QQQQ, KKKK, AAAA, 2222

(14) Rocket (1 action): BR

(15) Pass (1 action): PASS