# OpenReview forum: "PerfectDou: Dominating DouDizhu with Perfect Information Distillation"
_NeurIPS.cc/2022/Conference — NeurIPS 2022 Accept_

### Official Review · Reviewer_3trj · 2022-07-06

**Rating:** 6
**Confidence:** 4
**Soundness:** 3 good
**Presentation:** 2 fair
**Contribution:** 2 fair

**Summary:**

This paper presents PerfectDou, a novel AI agent for the game of PerfectDou.  This agent uses a training paradigm introduced by the authors called Perfect Training Imperfect Execution (PTIE) where in an actor-critic framework the critic has access to the exact state of the game, including all hidden information, while the actor/policy only has access to the imperfect information that players normally have access to.  The networks representing the actor/critic are trained using PPO with GAE.  The paper details the features used as input to the networks, additional reward information that is included to guide the policy.  The PerfectDou agents is then evaluated against other published DouDizhu agents, including the current state-of-the-art, DouZero. These results demonstrate that PerfectDou can beat these previous agents at the game and should now be considered the state of the art agent at the game.


**Questions:**

- Could you clarify the explanation of the result with high WP and negative ADP again?  I didn't follow the explanation on lines 311-315.
- It wasn't clear to me what matters more, winning or advantage points.   In a typical single game, is the goal simply to win, or does the reward (like the size of the pot in poker) actually depend on the advantage of the players at the end?  Or is it important in a series of games where the scores for the players are added up until someone gets to a certain score total, or something? And this work is just focused on a single game of DouDizhu?  It seems like the true primary goal should be winning, and the advantage only measures how close it was.  I believe though, it has been observed with other superhuman AIs, like in Go, that they play to win, and often the best chance of winning is accompanied by a very small winning margin in the end. In your case, the RewardlessDou has a better WP against DouZero(~1e10) than PerfectDou.  If WP is most important, than the node rewards appear unnecessary.
- Are there principles you used in your feature engineering that would be applicable in other, different domains?
- In what domains would you expect the PTIE paradigm to be similarly effective? Any imperfect-information setting? Is cooperation required? Multi-players (beyond 2)?


**Limitations:**

The main limitation that stands out to me is that this work assumes that it is playing with its same agent in the other peasant seat.  One of my big questions is how well the agents do in other pairings.  For example, if one DouZero peasant is paired with one PerfectDou peasant against another agent, how well does this work?  I understand that this is perhaps beyond the scope of this work, but I think that it should be made clear in the paper itself. I don't see any potential negative societal impact that should have been addressed.


**Strengths And Weaknesses:**

Strengths -
- The domain in question, DouDizhu, is an interesting one combining many interesting characteristics: competition, cooperation, and imperfect-information.  The success of this agent represents an advancement in these areas and I think this is significant enough to warrant sharing with the broader AI/ML community.
- To my knowledge, the perfect-training-imperfect-execution (PTIE) paradigm introduced by the authors is novel and the experimental results demonstrate that it contributes to the success of the agent. This paradigm could be useful in other similar imperfect-information settings.
- The high-level presentation of the methods is generally clear, with a few execeptions noted below.
- It is clear that the underlying work was carefully and thoroughly done, and is of high quality.

Weaknesses:
- The paper is singularly focused on the domain of DouDizhu. It remains up to the reader to imagine how any of the insights from this work might translate to other domains and problems. I think that the work would be strengthened if it included some of this discussion in the paper itself.
- With the exception of PTIE, this largely appears to be an engineering feat specific to DouDizhu, and it is unclear how many contributions are significant elsewhere.  All of the algorithms and methods appear to be drawn from previous work, and while there is value in generating this productive combination for this domain, it could be stronger.
- Ablation study - If I am understanding correctly, Vanilla PPO doesn't have access to the perfect information features or the node reward.  What is this testing exactly?
- The low-level clarity of the writing was poor at times, and there are several sentences (noted below) where I couldn't determine what was being communicated.


Minor Feedback:
- Line 41 - "when deploy the..." should be "when deploying the..."
- Line 143-144 - "In addition, the optimality of RL is .... " - This sentence doesn't make sense to me.
- Line 187 - should read "... the imperfect features are a subset of the perfect features."
- Line 257-258 - you state that CFR is not appropriate for DouDizhu due (in part) to the "mixed-game property".  There is at least one example that I am aware of where CFR was used for this type of mixed-game setting.  In [Mazrooei, Parisa, Christopher Archibald, and Michael Bowling. 2013. “Automating Collusion Detection in Sequential Games”. Proceedings of the AAAI Conference on Artificial Intelligence] CFR was used to train players in 3 player poker, and two of those players were trained to collude/coordinate with each other, sharing their reward.  CFR was able to train successful agents in this case.
- Line 298 - You state that games are randomly generated and played twice, once with each algorithm controlling the landlord, and once controlling the two peasants.  I assume that the landlords get the same hand in each case?  Did you consider playing each game 6 times, once for each possible landlord hand (3) and each algorithm (2)? It seems this could reduce the variance of the game that you refer to on line 303.
- Lines 316-327 - What do you mean by frames?  Do you mean training iterations? or epochs?
- Line 357 - What type of machine were the inference times computed on?
- Lines 357-259 - "The inference time of each AI could be attributed to its solution and implementation in the playing time."  I don't understand what this sentence is saying.
- Line 374 - What does it mean for a player to be "leaky and unreasonable"?
- Lines 380- 389 - What is a "bomb" in the context of DouDizhu?
- Line 381-382 - "...and tends to control the game even the Peasants play more bombs;" should this be: "... and tends to control the game even when the Peasants play more bombs"?
- Line 386 - What do you mean by "more aggressive and less thinking" exactly?  Similarly for "guessing and supressing", "better at card combination" and "more calm" on the next lines.

---

> ### Author Response · Authors · 2022-08-01
> **Thanks for your careful check and detailed feedback!**
>
> **Q1: Lines 380- 389 - What is a "bomb" in the context of DouDizhu?**
>
> A1: Yeah this is a special term of DouDizhu and we have a brief introduction of 'bomb' in Appendix B.1 Line 577. In short, the bomb is a kind of card's category, such as 3333, 4444 or 5555, this kind of action is special because it suppresses all other kinds of categories (except rocket) and it will double the score. Playing the bomb will bring the privilege by suppressing the opponent's action while it also brings the risk of losing more scores if the player's other cards are not strong. It is necessary for the experienced player to find out the balance.
>
> **Q2: Could you clarify the explanation of the result with high WP and negative ADP again? I didn't follow the explanation on lines 311-315.**
>
> A2: Sorry for the confusion. A potential reason for high WP and negative ADP is that the agent is reckless to play out the bigger cards (like bombs) without considering the left hand. For example, consider the case when **AI1** always chooses to suppress by playing out its bomb and the opponent **AI2** only plays bomb when its left cards are good. In that case, if the left hand cards of **AI1** are similar or better than **AI2**, **AI1** can win most games with small scores. However, in the opposite scenario, where **AI2**'s left cards in hand are better than **AI1**, **AI1** and **AI2** both play out the bombs, making the score be doubled several times and **AI1** would lose plenty of scores. This could be one example of high WP and negative ADP caused by the ai player's wrong strategy. This can be further clarified in the revision.
>
> **Q3: It wasn't clear to me what matters more, winning or advantage points. In a typical single game, is the goal simply to win, or does the reward (like the size of the pot in poker) actually depend on the advantage of the players at the end? Or is it important in a series of games where the scores for the players are added up until someone gets to a certain score total, or something?**
>
> A3: Sorry for your confusion about the rules. As explained in Line 287-289 and Line 579-588, when human players play DouDizhu the Average Difference in Points (ADP) is more important than the winning percentage (WP). The goal of this game is to achieve as many scores as possible. In the real competition of DouDizhu, a series of games are played and the scores of players are added up, after all games end, the player with the highest score wins the game.
>
> **Q4: And this work is just focused on a single game of DouDizhu? It seems like the true primary goal should be winning, and the advantage only measures how close it was. I believe though, it has been observed with other superhuman AIs, like in Go, that they play to win, and often the best chance of winning is accompanied by a very small winning margin in the end.**
>
> A4: The answer to this question actually relates to the scoring rule of the game. In our work, we focused on building an AI which could play real DouDizhu competition with human players, where ADP is much more important, as mentioned above (Q2A2). Normally, the base score of the game is 2 for the landlord and 1 for each peasant, and when the landlord loses the game, it loses 2 scores and vice versa. However, because of the existence of `Bomb`, the score can be doubled. Some strategies like not playing bombs when the ranks of other cards in hand are small can avoid losing large scores.
>
> **Q5: In your case, the RewardlessDou has a better WP against DouZero(~1e10) than PerfectDou. If WP is most important, then the node rewards appear unnecessary.**
>
> A5: As explained above, from the main point of this paper, WP is **less important** than ADP since we want to win higher scores in series of games. Additionally, in our experiments, all AI systems are trained to maximize ADP by taking it as the basic reward (as stated in Line291).
> Furthermore, there is another evidence from Line 314 to 315 in the paper showing the reasonable range of WP value when we only want to maximize ADP, where we analyzed the statistics of online human matches and showed that the WP of the human player is usually kept in a range of 0.52 ∼ 0.55 when the player tries to maximize ADP.
>
> **Q6: Are there principles you used in your feature engineering that would be applicable in other, different domains?**
>
> A6: Thanks for the good question! We believe part of the features in card representation matrix (Figure 2) such as the number of cards in hand can be easily transferred to other games such as Mahjong or Texas Hold'em. For the node representation, we believe most of the features such as current cards in hand, last moves and cards in the opponent's hand are common features in card games and can be applied to other games as well.

---

> > ### Author Response · Authors · 2022-08-01
> > **Response cont.**
> >
> > **Q7: In what domains would you expect the PTIE paradigm to be similarly effective? Any imperfect-information setting? Is cooperation required? Multi-players (beyond 2)?**
> >
> > A7: We expect the PTIE paradigm is suitable for all imperfect-information games where Actor-Critic algorithms can be applied, and it is not limited to games featured with cooperation or multi-players since the aim is to utilize the global information.
> >
> > **Q8: You state that games are randomly generated and played twice ... Did you consider playing each game 6 times, once for each possible landlord hand (3) and each algorithm (2)?**
> >
> > A8: Thanks for the advice! In our submitted version, we mainly follow the setting of DouZero to make a fair comparison. In the revision, we have supplemented the extra experiments following your suggestions and the result is as follows:
> >
> > | A \ B | PerfectDou  | PerfectDou  | DouZero  | DouZero |
> > |---|---|---|---|---|
> > |   | WP | ADP  | WP  | ADP  |
> > | PerfectDou  | - | - | 0.544  | 0.150 |
> > | DouZero  |0.456 | -0.150 |  - | -  |
> >
> > **Q9: Lines 316-327 - What do you mean by frames? Do you mean training iterations? or epochs**
> >
> > A9: We are sorry for the confusion. The term `frames` mainly follows the common usage in the RL literature (also see in [1,2]). In fact, it simply means game frames, in terms of RL words, also known as observations or interaction steps. Therefore `1e9 and 2.5e9 frames` means we trained the network after obtaining 1e9 and 2.5e9 steps of observations. We have revised it as `steps` in our revision to make it clear.
> >
> > [1] Mnih V, Kavukcuoglu K, Silver D, et al. Playing atari with deep reinforcement learning[J]. arXiv preprint arXiv:1312.5602, 2013.
> > [2] Yarats D, Fergus R, Lazaric A, et al. Mastering visual continuous control: Improved data-augmented reinforcement learning[J]. arXiv preprint arXiv:2107.09645, 2021.
> >
> > **Q10: What type of machine were the inference times computed on?**
> >
> > A10: Sorry for missing such information! The inference times were computed on a single core of Intel(R) Xeon(R) Gold 6130 CPU @ 2.10GHz. We have now supplemented it into the main text.
> >
> > **Q11: The meaning of "The inference time of each AI could be attributed to its solution and implementation in the playing time"**
> >
> > A11: In this sentence, we want to explain the reason why the inference time of each AI varies. As explained in lines 359-374, the result is caused by their different solutions and implementations.
> >
> > **Q12: What does it mean for a player to be "leaky and unreasonable"? / What do you mean by "more aggressive and less thinking" exactly? /Similarly for "guessing and supressing", "better at card combination" and "more calm" on the next lines.**
> >
> > A12: We think the answers can be found in Appendix D.2 and D.3, where we show detailed qualitative comparisons between Douzero and PerfectDou, in-depth statistical analysis and a few detailed playing examples to give more evidence why PerfectDou can beat DouZero.
> >
> > **Q13: For example, if one DouZero peasant is paired with one PerfectDou peasant against another agent, how well does this work?**
> >
> > A13: Thanks for your great suggestion! This is a very interesting problem and unfortunately we missed it by simply following the competition setting of DouZero for fair comparisons. Extra experiments have been carried out in the revision (Appendix D.6) and the results of Landlord against Peasants are as following:
> >
> > |  Peasant →  | PerfectDou+DouZero | PerfectDou+DouZero | PerfectDou  | PerfectDou | DouZero | DouZero |
> > |---|---|---|---|---|---|---|
> > | **Landlord↓** | WP | ADP  | WP  | ADP  | WP | ADP  | WP | ADP  |
> > | **PerfectDou**  | 0.424 | -0.448 | 0.389 | -0.606 | 0.452 | -0.375  |
> > | **DouZero**  | 0.395 | -0.534 | 0.363 | -0.676 | 0.421 | -0.465 |
> >
> > In the experiment, two peasants are played by PerfectDou and DouZero respectively and the landlord is played by PerfectDou or DouZero, using 100,000 randomly generated games and each game is played 6 times as before.
> > Some conclusions could be drawn from the table: when playing as the peasant, PerfectDou can better cooperate with its teammate than DouZero. And when playing as the landlord, PerfectDou outperforms DouZero against all types of opponents.
> >
> > **Q14: Ablation study - If I am understanding correctly, Vanilla PPO doesn't have access to the perfect information features or the node. What is this testing exactly?**
> >
> > A14: One of the main differences between PerfectDou and DouZero is that DouZero is a value-based method while PerfectDou utilized an actor-critic algorithm - PPO, this experiment aimed to illustrate the improvement of PerfectDou vs DouZero is not arisen from simply replacing the underlying algorithm as PPO, the node reward and the PTIE both play significant roles.

---

> ### Author Response · Authors · 2022-08-04
> **All minor feedback has been responsed or revised in the paper**
>
> Hi dear Reviewer 3trj,
>
> We have further revised our manuscript following your detailed feedback and uploaded the revised version. Please let us know if you have more questions or discussions for us! Thanks again!
>
> The authors.

---

### Official Review · Reviewer_egmk · 2022-07-11

**Rating:** 6
**Confidence:** 4
**Soundness:** 4 excellent
**Presentation:** 3 good
**Contribution:** 3 good

**Summary:**

This paper proposes a SoTA DouDizhu AI system named PerfectDou that dominates the game. They adopt a perfect-training-imperfect-execution framework. It uses the global information to guide the training process. During the actual gameplay, only imperfect information can be used. Furthermore, they also do some feature engineering on card and action representations. When training the AI agent, they adopt PPQ with GAE by self-play in a parallel training system. Finally, extensive experimental results show that PerfectDou can achieve SoTA performance.

**Questions:**

1.	Is the perfect-information-imperfect-execution training paradigm a general structure for imperfect information games? In other words, can it be used for solving other imperfect-information games, such as other poker games?
2.	Does it just train one policy network for both peasants and a landlord? Or does it train a policy network for each player?


**Limitations:**

This paper does not claim the limitations and potential negative societal impact. Maybe the decision in the bidding phase needs to be considered in the future.

**Strengths And Weaknesses:**

Strengths: The idea of perfect-information-imperfect-execution paradigm for training imperfect-information games is interesting. It can effectively use the perfect information for training and distill it into policy network, which can only use imperfect information. To further improve the performance of DouDizhu AI agent, they also characterize card and game features to represent the perfect and imperfect information. These may be the main contributions of this paper. The structure of the paper is well, and this paper is easy to follow. This paper indeed provides a novel method for games with competition and collaboration.

Weaknesses: This paper only focuses on the card play phase. The strategy in the bidding phase may also influence the whole performance of the AI agent. Another slight problem is inference time. As the experimental results show, although the inference time of PerfectDou is much fast, the inference time is slower than DouZero.

---

> ### Author Response · Authors · 2022-08-01
> **Thanks for your valuable comments!**
>
> **Q1: This paper only focuses on the card play phase. The strategy in the bidding phase may also follows influence the whole performance of the AI agent.**
>
> A1: Random selection is used in the bidding phase following the setting of DouZero to make a fair comparison. In practice, a bidding network used in the bidding phase is trained to predict the winning probability with the initial hand.
>
> **Q2: Inference time concern.**
>
> A2: The inference time consists of two parts, feature processing procedure and model inference. We test that the network inference time of DouZero is 2.2 ms(pytorch) and the feature extraction time is 0.39 ms while the network inference time of PerfectDou(tensorflow) is 5.0 ms and the feature extraction time is 3.6 ms. As stated in Line 370-371, PerfectDou is a bit slower than DouZero due to the more complex feature processing procedure and model structure, which can be largely optimized into neglectable computation cost from practical implementation, such as changing the source code from Python to C++, which is common for AI deployment.
>
> **Q3: "Is the perfect-information-imperfect-execution training paradigm a general structure for imperfect information games? In other words, can it be used for solving other imperfect-information games, such as other poker games?"**
>
> A3: We are sorry for not including such a discussion in the main text, more discussions are given in Appendix F in the revised version, which will be further supplemented into the main text in the future version. In this paper, due to the amount of work, we mainly focus on one of hard poker games, DouDizhu, but we surely believe the technique will boost AI works on other imperfect-information games such as Texas Hold'em or Mahjong due to the success of utilizing global information in many multi-agent challenges.
>
> **Q4: Does it just train one policy network for both peasants and a landlord? Or does it train a policy network for each player?**
>
> A4: This is covered in the paper. As stated in Appendix D.1, Line 611-613, "During self-play, we find a better practical solution for DouDizhu is to keep three different models for Landlord and two Peasants separately which is only updated by their own data against the latest opponent model." The three models mean Landlord, Peasant before Landlord and Peasant after Landlord.

---

### Official Review · Reviewer_jTJJ · 2022-07-11

**Rating:** 6
**Confidence:** 3
**Soundness:** 3 good
**Presentation:** 3 good
**Contribution:** 2 fair

**Summary:**

This paper mainly focuses on the DoDizhu Game. It proposes the Perfect-Training-Imperfect-Execution(PTIE) learning framework to train a robust agent called PerfectDou. Specifically, under the actor-critic framework, it uses imperfect information to feed the actor and uses perfect information to feed the critic. Combined with the new representation (Card, Node, and Action) and reward design, it trains the agent which can beat previous AI on DoDizhu and achieve the SOTA performance. In addition, they conduct ablation experiments to verify that the PTIE framework, feature design, and node reward all contribute to the final performance.

**Questions:**

1. I do not see much difference between the perfect-training-imperfect execution (PTIE) paradigm and the centralized-training-decentralized-execution. Both of them seem to use the global state to train the critic and partial observation to train the actor.

2. I doubt if the PTIE framework plays an important role in this work. I wonder if the Douzero gets better performance if it also combines the feature design and the node reward.

**Ethics Review Area:**

["I don’t know"]

**Limitations:**

I think this work has no potential negative societal impact.

**Strengths And Weaknesses:**

Originality
This work proposes the three main parts: Perfect-Training-Imperfect-Execution (PTIE) learning framework, a new representation on card, node, and action, and a new node reward design. Combining the three parts can train the AI with SOTA performance in DouDizhu game. However, I do not see much difference between the PTIE and the CTDE, and the latter two are more dependent on the human experience. So, I think the originality of this paper is limited.


Quality
From an engineering perspective, I think this is a good work.

Clarity
I think this paper is written clearly.

Significance
The AI trained by the method proposed in this work can achieve the SOTA performance in the Doudizhu game. However, this framework may only be applicable to Doudizhu because the feature and reward design play a vital role, but they can only be used in the Doudizhu game.

---

> ### Author Response · Authors · 2022-08-01
> **Thanks for your critique and positive feedback!**
>
> **Q1: no much difference between PTIE and CTDE.**
>
> A1: Essentially, we think you are right. L38-39 has claimed that the "Perfect-Training-Imperfect-Execution (PTIE) framework, a variant of the popular Centralized-Training-Decentralized-Execution (CTDE) paradigm". Although they both utilize global information, technically, the main difference is their form of global information. In detail, most of previous CTDE works focus on allocating the observations and actions across agents; by contrast, PTIE concentrates on the range of the information itself in imperfect-information games, which does not make any requirement on agents' joint observations or actions.
>
> As stated in Line 532 Appendix A, where we further discussed related works of training with global information, we are the first to formulate the idea for training agents on imperfect-information games, making the idea explicitly useful for more kinds of imperfect-information games.
>
> **Q2: "I doubt if the PTIE framework plays an important role in this work. I wonder if the Douzero gets better performance if it also combines the feature design and the node reward."**
>
> A2: The concern is already addressed in the ablation studies, Table 4. In Line 335-338, we design the battle among **ImperfectDouZero** (DouZero with our proposed imperfect-information features) and **ImperfectDou** (PerfectDou with only imperfect features as inputs for the value function.) and **RewardlessDou** (PerfectDou without node reward) and even **Vanilla PPO** (Naive actor-critic training with imperfect-features only and without additional reward). We try to disentangle all important components in our proposed method to validate the effectiveness of each part.
>
> For your question, the analysis can be referred to the battles between **ImperfectDouZero v.s. RewardlessDou** and **ImperfectDouZero v.s. Vanilla PPO**. Note that DouZero cannot use the perfect information since it will play in a cheating style if doing so (it is a value-based method thus it only has the value function and no policy function to distill). ImperfectDouZero uses the same imperfect features of PerfectDou, RewardlessDou does not include the node reward and Vanilla PPO does not include the node reward and PTIE. From Table 4, we can conclude that compared with Vanilla PPO, RewardlessDou shows a significant improvement under the help of PTIE when playing against ImperfectDouZero.
>
> The above analysis will be further revised and supplemented into the final version.

---

### Author Response · Authors · 2022-08-08
**We sincerely look forward to your reply.**

Dear reviewers,

We first thank you again for your valuable comments and suggestions. In the previous replies, we have tried our best to address your questions point by point and supplemented more experiments.

We sincerely look forward to your reply to our response. And we are open to any discussion to improve our paper.

Best wishes!

The authors.

---

### Meta-Review · Area_Chair_aut3 · 2022-08-25

**Recommendation:** Accept
**Confidence:** Certain

**Metareview:**

The reviewers appreciate both main contributions, namely the PTIE concept and the feature and reward engineering. While there are concerns that neither may generalize beyond the specific game of DouDizhu, and that PTIE may be somewhat incremental given CTDE (or even not novel at all; several CTDE works use the entire state for the centralized training), successfully demonstrating PTIE on even this single domain and methodically evaluating its effect should be of interest to the community. The paper is mostly well written, and the authors are requested to incorporate the specific reviewer feedback.

**Award:**

No

---

### Decision · Program_Chairs · 2022-09-14

Accept